# Intrinsic factors and CD1d1 but not CD1d2 expression levels control invariant natural killer T cell subset differentiation

Ludivine Amable[1], Luis Antonio Ferreira Martins[1], Remi Pierre[2], Marcio Do Cruseiro[2], Ghita Chabab[3], Arnauld Sergé [4], Camille Kergaravat[1], Marc Delord[5], Christophe Viret[6], Jean Jaubert[7], Chaohong Liu[8], Saoussen Karray [9], Julien C. Marie [3], Magali Irla [10], Hristo Georgiev [11], Emmanuel Clave [1], Antoine Toubert [1], Bruno Lucas[12], Jihene Klibi[1] & Kamel Benlagha [1] ✉

Invariant natural killer T (NKT) cell subsets are defined based on their cytokine-production profiles and transcription factors. Their distribution is different in C57BL/6 (B6) and BALB/c mice, with a bias for NKT1 and NKT2/NKT17 subsets, respectively. Here, we show that the non-classical class I-like major histocompatibility complex CD1 molecules CD1d2, expressed in BALB/c and not in B6 mice, could not account for this difference. We find however that NKT cell subset distribution is intrinsic to bone marrow derived NKT cells, regardless of syngeneic CD1d-ligand recognition, and that multiple intrinsic factors are likely involved. Finally, we find that CD1d expression levels in combination with T cell antigen receptor signal strength could also influence NKT cell distribution and function. Overall, this study indicates that CD1d-mediated TCR signals and other intrinsic signals integrate to influence strain-specific NKT cell differentiation programs and subset distributions.

Invariant Natural Killer T (NKT) cells recognize lipid antigens presented by CD1d, a non-classical class I-like major histocompatibility complex (MHC) molecule. These cells express a semi-invariant T cell antigen receptor (TCR) made up of an invariant TCRα chain (Vα14-Jα18 in mice, Vα24-Jα18 in humans) associated mainly with TCRβ8.2, Vβ7, or Vβ2 in mice, and Vβ11 in humans[1,2]. During development, NKT cells acquire effector/memory features linked to the key transcription factor promyelocytic leukemia zinc finger (PLZF), which is also important for mucosal-associated invariant T (MAIT) and γδ T cell development[3–5]. In the thymus, NKT cells develop into three major terminally differentiated and functionally distinct NKT cell subsets. NKT1 cells express the transcription factor T-bet and predominantly secrete IFNγ; NKT2 cells express high levels of GATA3 and PLZF transcription factors and secrete IL-4 and IL-13; NKT17 have intermediate levels of PLZF, are positive for RAR-related orphan receptor gamma (RORγt), and secrete IL-17[6].

[1]Université Paris Cité, Institut de Recherche Saint-Louis (IRSL), EMiLy, Paris, France. [2]Plateforme de recombinaison homologue et de cryoconservation (PRHTEC), Institut Cochin, Université Paris Descartes, Paris, France. [3]Tumor Escape Resistance and Immunity department, Cancer Research Center of Lyon INSERM U1052, CNRS UMR 5286, Centre Léon Bérard, Lyon, France. [4]Laboratoire Adhésion Inflammation (LAI), CNRS, INSERM, Aix-Marseille Université, Marseille, France. [5]King's College London, London, UK. [6]CIRI, Centre International de Recherche en Infectiologie, Université de Lyon, INSERM U1111, Université Claude Bernard Lyon 1, CNRS, UMR5308, ENS de Lyon, Lyon, France. [7]Mouse Genetics Unit, Institut Pasteur, Paris, France. [8]Department of Pathogen Biology, School of Basic Medicine, Tongji Medical College, Huazhong University of Science Technology, Wuhan, China. [9]Université Paris Cité, INSERM U976, Institut de Recherche Saint-Louis (IRSL), Hôpital Saint-Louis, Paris, France. [10]Centre d'Immunologie de Marseille-Luminy (CIML), CNRS, INSERM, Aix-Marseille Université, Marseille, France. [11]Institute of immunology, Hannover Medical School, Hannover, Germany. [12]Institut Cochin, Centre National de la Recherche Scientifique UMR8104, INSERM U1016, Université Paris Descartes, Paris, France. ✉e-mail: kamel.benlagha@inserm.fr

NKT cells develop in the thymus from CD4 + CD8+ double-positive (DP) thymocytes expressing a Vα14-Jα18 TCR recognizing glycolipid antigens. These antigens are presented by CD1d on neighboring DP thymocytes in a SLAM-SAP-mediated co-stimulation context[7]. NKT cell development can be subdivided into the following four stages: 0 (HSA + CD44- NK1.1-), 1 (HSA- CD44- NK1.1-), 2 (HSA- CD44 + NK1.1-), and 3 (HSA- CD44 + NK1.1+)[8]. The closest cells to positive selection, stage 0, express HSA, CD69, and Egr-2. They represent the earliest common NKT cell precursor[9,10]. Selection of these cells is believed to involve a strong TCR agonist positive selection signal triggering the expression of Egr-2/PLZF, and entry of the cell into the NKT cell program[11,12]. Our previous work suggests that high TCR density might contribute to TCR signal strength at stage 0[13]. Stage 1 then comprises the NKT progenitor population, known as NKTp[14]. These cells are identified based on the upregulation of PLZF and acquisition of CCR7 and S1PR1. Slamf6, and to a lesser extent Slamf1, are required for iNKT cell development in mice[7]. However, whether SLAM-SAP (SLAM-associate proteins) mediated co-stimulation positively or negatively regulates TCR signaling in NKT cells remains unclear. In vitro co-stimulation of Slamf6 and TCR with immobilized antibodies was shown to potentiate Egr2 and PLZF expression in murine pre-selection DP thymocytes[15–17]. However, deletion of the entire slam family receptor (SFR) locus in mice led to reduced NKT cell populations as a consequence of increased TCR signaling and reduced survival, suggesting that SFRs promote NKT cell development by reducing TCR signaling[18,19]. Thus, how TCR and SFR signals integrate to control NKT cell selection and effector differentiation remains an unresolved key question.

TCR-mediated effector subset instruction could occur during cortical selection and/or in the cortex or medulla following the NKTp stage. Various studies have shown that CD1d/TCR interaction is important for the functional maturation of NKT cells[20–22]. In fact, stage 1 and 2 NK1.1- NKT cells progressed to stage 3 NK1.1+ following intrathymic injection in WT, but not CD1d-KO mice[20,21]. Our group further showed that cortical thymocytes along with their selecting ligand are required for the NKT cell maturation to stage 3[20]. CD1d/TCR interaction was also found to regulate NKT2 function as mature thymic NKT2 cells that normally produce IL-4 at the steady state no longer do so in a CD1d-deficient environment[22]. A hierarchy of TCR expression levels and signal strengths in NKT cell subsets has been established (NKT2 > NKT17 > NKT1)[17]. This ranking led to the hypothesis that TCR signal strength might influence the fate of developing NKT cells. This hypothesis was explored in two studies using mouse models expressing hypomorphic alleles of *Zap70* – SKG BALB/c mice, YYAA and ZAP70AS B6 mice[17,23]. NKT1 cells were present at normal levels in these animals, but the frequency of thymic NKT2 and NKT17 cells was decreased, indicating that the development of these subsets requires stronger TCR signals than NKT1 cell development[17,23]. Further evidence of a role for TCR signaling strength in NKT differentiation was provided by B6 mice lacking a negative regulator of TCR signals – the enzyme A20 (also known as TNFAIP3). In these mice, decreased NKT1 cell numbers were observed in the thymus and peripheral tissues[24]. To systematically investigate the earliest phases of NKT cell development and subsequent subset differentiation, a TCR-inducible mouse model (CD4–creERT2 Vα14iStopF Nur77– eGFP) was developed by the Schmidt-Supprian group[25]. Using this model, NKT cell development can be generated in an inducible wave, making it possible to define the precise temporal sequence of events guiding NKT cell development. Results obtained with this model revealed that early TCR signaling instructs a common progenitor state encompassing NKT0 cells, with effector subsets emerging later without further TCR input. Evidence was also provided that NKT17 cells derive from these progenitors within 3 to 5 days over the course of a short proliferative phase. NKT1 cells, in contrast, emerge continuously, differentiating from NKTp/NKT2-like cells over 14–20 days probably through a cytokine-driven process. Contrasting results were obtained using mouse models expressing hypomorphic *Zap70* alleles (SKG and YYAA)[17,23,24]. In these models, differences in TCR expression levels and signaling were not obvious at the NKT0 cell stage, and only emerged at subsequent developmental stages[17]. Thus, how and when TCR signaling control NKT cell selection and effector differentiation is still a matter of debate.

In mice, two genes arranged in opposite transcriptional orientation have been identified, that encode the CD1d isoform: *Cd1d1* and *Cd1d2*[26]. The *Cd1d2* gene in B6 mice contains a frame shift mutation at the beginning of the fourth exon, coding for the α3 domain. This mutation abolishes CD1d2 protein expression in this strain. In BALB/c mice, the reading frame for *Cd1d2* allows its transcription, and potentially its expression at the cell surface[27]. A *Cd1d2* transgenic mouse has been successfully generated on a B6 CD1d−/− background[28]. However, due to the very low levels of CD1d2 expression on the surface of cortical thymocytes – necessary for NKT cell selection – this mouse strain nevertheless lacks NKT cells. The authors of the study proposed that the low expression levels could be explained by a transcriptional deficiency. In BALB/c mice, CD1d1 and CD1d2 mRNAs are expressed at equivalent levels, suggesting that the corresponding proteins will be expressed at similar levels at the cell surface. However, as no antibodies discriminating between CD1d1 and CD1d2 molecules exist, expression of CD1d2 has never been formally documented on the surface of BALB/c thymocytes.

Importantly, CD1d2 molecules have been shown to present short lipid antigen chains to NKT cells, indicating that they can contribute to positive selection of NKT cells in the same way as CD1d1 molecules, which present long lipid chains[29]. A recent analysis of mice expressing CD1d2 on a B6 CD1d1 KO background showed that the resulting repertoire differs to that selected by CD1d1, being more NKT2/NKT17-skewed[29].

The relative distribution of the three thymic NKT subsets – NKT1, NKT2, and NKT17 – varies depending on the mouse strain, and affects the phenotype and activation status of surrounding cells[6]. In several mouse strains, including BALB/c, NKT2 cells are abundant, and following stimulation by self-ligand, produce IL-4. In these strains, IL-4 conditions CD8 T cells to become "memory-like", triggers production of chemokines by thymic dendritic cells, and thymic egress of mature "conventional" T cells[6,30,31]. Additionally, RANKL-expressing thymic NKT2 and NKT17 cells regulate the differentiation of Aire+ MHC class II + medullary thymic epithelial cells, which are involved in clonal deletion of self-reactive T cells and maturation of Treg[32]. Thymus-resident NKT1 cells produce IFN-γ and promote Qa2 expression in CD4 and CD8 thymocytes, thus contributing alongside TNF-α to the maturational process for these cells[14].

As mentioned above, differential TCR signal strength perceived by NKT cells has been proposed to direct subset differentiation, with NKT2 > NKT17 > NKT1[17,23]. We therefore hypothesized that differences in NKT cell subset distribution between strains could be linked to TCR signal strength. To test this hypothesis, we compared TCR signal strength in NKT cell from B6 and BALB/c mice. In BALB/c mice, we found NKT cells, and particularly NKT2 cells, to perceive a higher TCR signal strength compared to B6 mice. This difference was not due to a higher CD1d or SLAM expression in BALB/c mice or to differences in CD1d1/CD1d2 isomorph expression as CD1d2 was poorly expressed in BALB/c mice and was inefficient for NKT cell selection. Using mixed bone marrow chimeric mice, with CD1d1−/−CD1d2−/− B6 or CD1d1−/−CD1d2−/− BALB/c NKT cells developing on CD1-expressing WT BALB/c or B6 cortical thymocytes, respectively, we found that NKT cells from the CD1d1−/−CD1d2−/− compartment distribute similarly to their strain of origin. Finally, analysis of NKT cell development in CD1d+/− mice showed that CD1d expression levels could influence NKT cell subset distribution and function.

## Results

### TCR signal strength correlates with specific NKT cell subset composition

We first confirmed that the reported NKT cell subsets[6] were present in the relevant proportions in our B6 and BALB/c mice. As expected, lower frequencies and numbers were observed in B6 compared to BALB/c mice (Fig. 1a, and Supplementary Fig. 1a). In addition, NKT1 cells represented the major NKT cell subset in B6 mice, whereas NKT2 cells predominated in BALB/c mice (Fig. 1a). These two mouse strains are therefore suitable to decipher factors driving strain-specific selection and differentiation.

Several reports have investigated how TCR signal strength influences NKT cell subset differentiation[17,23,24]. Therefore, we next analyzed CD5, PLZF, and Egr2 expression levels in NKT cells in the two strains as an indication of TCR signal strength. These markers were expressed at lower levels on total NKT cells from B6 mice (Fig. 1b), and mainly present on NKT2 cells (Supplementary Fig. 1b). We verified that NKT cells in BALB/c mice were of similar size, meaning there was no need to correct MFI for a cell size difference (Supplementary Fig. 1c). These results confirmed that NKT cells in these mice would perceive a lower TCR signal strength. The reduced expression of these markers in NKT cells from B6 mice correlated with lower TCR and SLAM expression on total NKT cells (Fig. 1c) and on all NKT cells subsets (Supplementary Fig. 1d) compared to BALB/c mice. In contrast, CD1d and SLAM expression on cortical thymocytes was higher in B6 compared to BALB/c mice (Fig. 1d). These results indicate a correlation between NKT cell subset composition and TCR signal strength in B6 mice that differs from the situation in BALB/c mice. They also support the notion that TCR signal strength could influence NKT cell subset differentiation.

### CD1d isomorph expression does not cause strain-specific NKT cell distribution

As CD1d expression levels on cortical thymocytes do not positively correlate with TCR signal strength, we considered whether CD1d isomorphs could cause the differential TCR signal strength and NKT cell distribution between B6 and BALB/c mice. As no serological reagent currently exists discriminating between the two proteins, their relative surface expression and contribution to NKT cell development is unknown. To address this question, we used Crispr-cas9 technology to generate BALB/c mice expressing either CD1d1 or CD1d2 molecules, introducing a reading frame shift creating an early stop codon in the relevant genes (Fig. 2a, and Supplementary Fig. 2). Analysis of NKT cell development in BALB/c CD1d2−/− mice, expressing CD1d1, showed a thymic NKT cell frequency, absolute numbers (Fig. 2b), and subset distribution (Fig. 2c, d) close to those observed in BALB/c littermate control mice. Similar analysis in BALB/c CD1d1−/− mice, expressing CD1d2, showed an approximately 8-fold reduction in thymic NKT cell frequency (6.5% vs 0.8%) and numbers ($10^6$ vs $1.2 \times 10^5$) (Fig. 2b), making the theoretical proportion of this population around 12% (100%: 8%) of NKT cells in BALB/c mice.

We also analyzed B6 CD1d1−/− mice that expressed CD1d2 as they were generated with ES cells from the 129/SV mouse strain. In these mice, NKT cell selection was even less efficient than in BALB/c CD1d1−/− mice: A 45-fold lower frequency than naïve B6 mice (0.08% vs 3.6%) was observed (Fig. 2e). NKT cell numbers are also dramatically reduced (Fig. 2e). Analysis of NKT cell subsets in BALB/c CD1d1−/− mice, expressing CD1d2, showed an increase in NKT1 and a decrease in NKT2 and NKT17 cell subsets compared to WT BALB/c littermate controls (Fig. 2c, d). This distribution profile resembles the one observed in B6 and B6 CD1d1−/− mice, expressing CD1d2 (Fig. 2f, g).

Analysis of CD1d1 expression on cortical thymocytes in BALB/c CD1d2−/− mice, expressing CD1d1, showed normal levels compared to WT BALB/c littermate controls (Fig. 3a, left histogram plot and

histogram). However, in BALB/c CD1d1−/− mice, expressing CD1d2, reduced CD1d2 surface expression (up to 30 times lower) was found in cortical thymocytes compared to CD1d1 expression in BALB/c CD1d2−/− (MFI: 100 vs 3000, respectively) or CD1d expression in WT BALB/c mice, expressing CD1d1 and CD1d2 (MFI: 100 vs 3000, respectively). These differences in intensity likely explain the low selection efficiency of NKT cells in these mice (Fig. 2b). In addition, CD1d2 was undetectable in the periphery of these mice (Fig. 3a, middle histogram plots and histograms). We observed the same trend for CD1d2 expression in B6 CD1d1−/− mice, expressing CD1d2, (Fig. 3b, left and middle histogram plot and histogram).

Overall, analysis of BALB/c mice indicated that CD1d1 and CD1d2 molecules promote the development of distinct NKT cell subsets: 1/ CD1d1 promotes the emergence of mostly NKT2/17 cells, inducing a "BALB/c-like" subset distribution in CD1d2−/− mice. 2/ CD1d2 promotes mostly NKT1 cell development inducing a "B6 like" subset distribution of NKT cells in CD1d1−/− mice (see Fig. 2d, g).

Despite this effect, CD1d2 is likely not the reason for the difference in subset distribution between B6 (mainly NKT1 cells) and BALB/c mice (mainly NKT2/17). This is due, first, to its low expression in DP thymocytes, and second, to the fact that CD1d2 mostly promotes NKT1 cell development.

In addition to showing that the CD1d isomorph could contribute to differentially promoting the development of NKT cell subsets, our results suggest that the genetic background could be determinant in this process. This is exemplified by the fact that mice expressing CD1d1 molecules in WT B6 (with a CD1d2 pseudogene) and BALB/c CD1d2−/− mice, expressing only CD1d1, did not have the same NKT cell subset distribution. Therefore, CD1d-mediated ligand recognition alone is not sufficient to promote a specific differentiation path.

### Factors intrinsic to hematopoietic cells contribute to the NKT cell subset differential distribution profile

To directly test whether CD1d-ligand recognition in the context of a B6 vs BALB/c genetic background plays a role in NKT cell subset differentiation, we investigated NKT cell development in mixed B6/ BALB/c chimeric mice. These experiments require mixed F1: B6 CD1d1−/−CD1d2−/− x BALB/c CD1d1−/−CD1d2 −/− recipient mice (hereafter referred to as F1: (B6 x BALB/c) CD1d1/2−/− recipient) to allow normal reconstitution. We first examined whether factors intrinsic or extrinsic to hematopoietic cells contributed to the differential NKT cell subset distribution profile in B6 and BALB/c mice. To do so, we transferred B6 (H-2Kb) or BALB/c (H-2Kd) BM cells into lethally-irradiated F1 (B6 x BALB/c) CD1d1/2−/− recipients. We analyzed the NKT cell subset composition of donor-derived NKT cells in the thymus and spleen 8-10 weeks post-BM reconstitution. B6 BM cells gave rise to high frequencies of NKT1 cells and BM BALB/c gave rise to high frequencies of and NKT2 cells in the thymus and spleen (Fig. 4a, left and right panel). These results corroborate the corresponding observations from wild type B6 and BALB/c mice. However, we also observed that BALB/c BM cells gave rise to a low frequency of NKT17 cells in the thymus and spleen, and to high frequencies of NKT1 cells, exceeding the NKT2 cell frequency only in the thymus (Fig. 4a, right panel). Importantly, a similar NKT cell subset distribution pattern was observed when BALB/c BM cells were grafted into BALB/c CD1d1−/−CD1d2−/− hosts (Fig. 4b, right panel). Consequently, the reversed NKT1/NKT17 distribution pattern observed in mice reconstituted with BALB/c BM cells is intrinsic to the grafted cells, and not extrinsically related to the F1: (B6 x BALB/c) CD1d1/2−/− recipient.

As expected, B6 BM cells grafted into B6 CD1d1−/− CD1d2−/− BM chimeras produced an NKT cell distribution resembling that observed in B6 mice (Fig. 4b, left panel). Hereafter, the thymic NKT cell distribution pattern observed following BALB/c BM reconstitution (high NKT2/NKT1 and low NKT17 frequency) will be considered as the

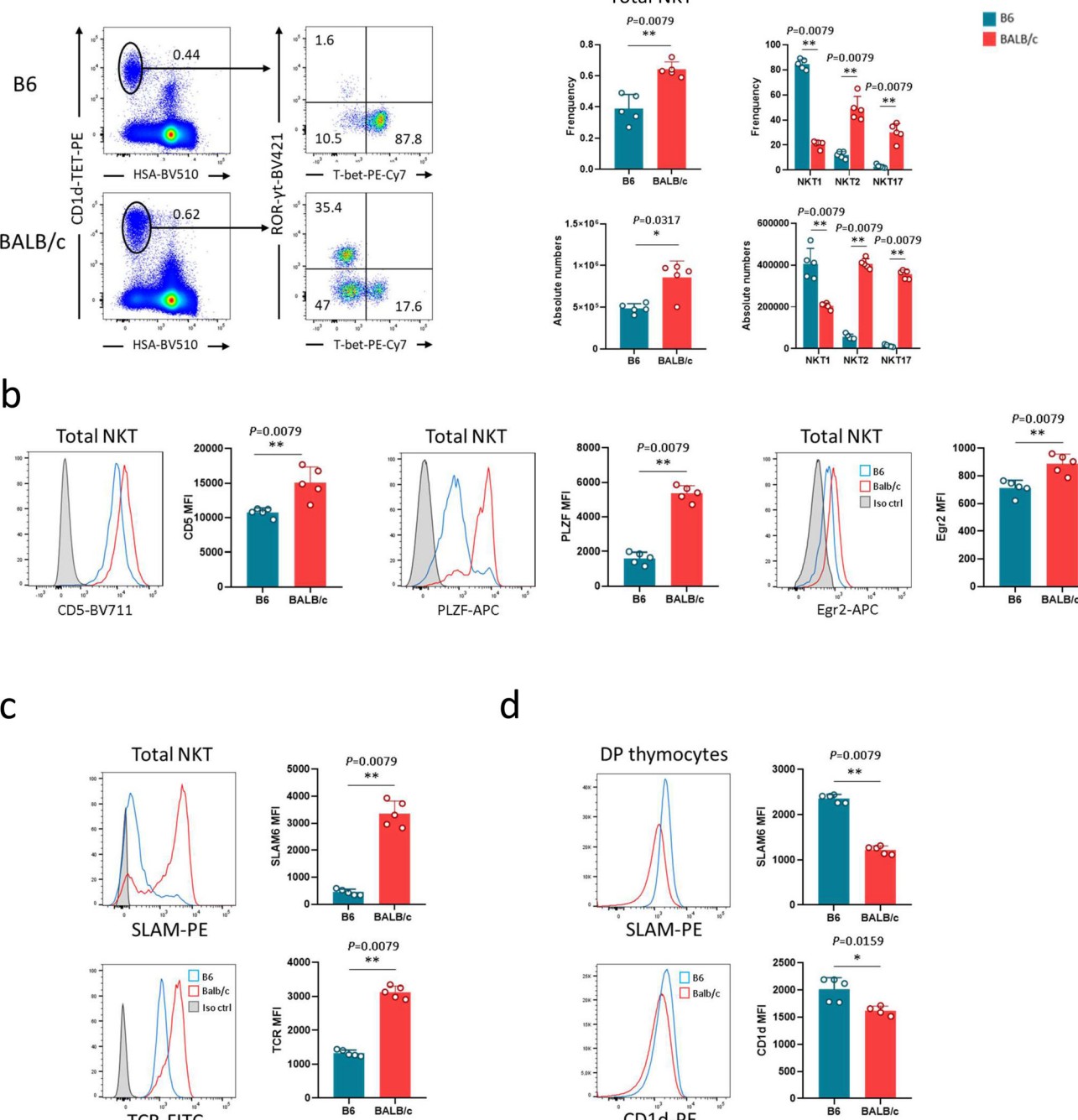

**Fig. 1 | TCR signal strength differs between B6 and BALB/c NKT cells.**
**a** Representative staining for NKT1 (T-bet[+]), NKT2 (T-bet[−]RORγt[−]), and NKT17 (RORγt[+]) subsets among thymic NKT cells from B6 (*n* = 5) and BALB/c (*n* = 5) mice. Numbers on dot plots correspond to frequencies. Individual/mean + SEM of mean frequency and absolute numbers for total NKT cells and each NKT cell subset are shown in the right panel. Statistics were calculated with the nonparametric Mann-Whitney test, two-sided, frequency B6 vs. BALB/c: **\*\***P = 0.0079; absolute numbers B6 vs. BALB/c: \*P = 0.0317; frequency NKT1, NKT2, and NKT17 B6 vs. BALB/c: \*\*P = 0.0079 for all subsets; absolute numbers NKT1, NKT2, and NKT17 B6 vs. BALB/c: \*\*P = 0.0079 for all subsets. **b** Representative staining for CD5, PLZF, and Egr2, in total thymic NKT cells from B6 (*n* = 5) and BALB/c (*n* = 5) mice. Individual/mean + SEM of mean MFI of these markers are shown to the right of each histogram plot. Statistics were calculated with the nonparametric Mann-Whitney test, two-sided,

frequency CD5, PLZF, and Egr2 MFI B6 vs. BALB/c: \*\*P = 0.0079 for all markers.
**c** Representative staining for SLAM6 and TCR in total thymic NKT cells from B6 (*n* = 5) and BALB/c (*n* = 5) mice. Individual/mean + SEM of mean MFI of these markers are shown in the right panel. Statistics were calculated with the nonparametric Mann-Whitney test, two-sided, frequency SLAM6 MFI B6 vs. BALB/c: \*\*P = 0.0079; TCR MFI B6 vs. BALB/c: \*\*P = 0.0079. **d.** Representative staining for SLAM6 and CD1d on cortical DP thymocytes from B6 (*n* = 5) and BALB/c (*n* = 5) mice. Individual/ mean + SEM of mean MFI of these markers are shown in the right panel. Statistics were calculated with the nonparametric Mann-Whitney test, two-sided, frequency SLAM6 MFI B6 vs. BALB/c: \*\*P = 0.0079; CD1d MFI B6 vs. BALB/c: \*P = 0.0159. Data are representative of 7, 5, 4, and 4 experiments in **a**–**d**, respectively, with 7-8-week-old mice. Source data are provided as a Source Data file.

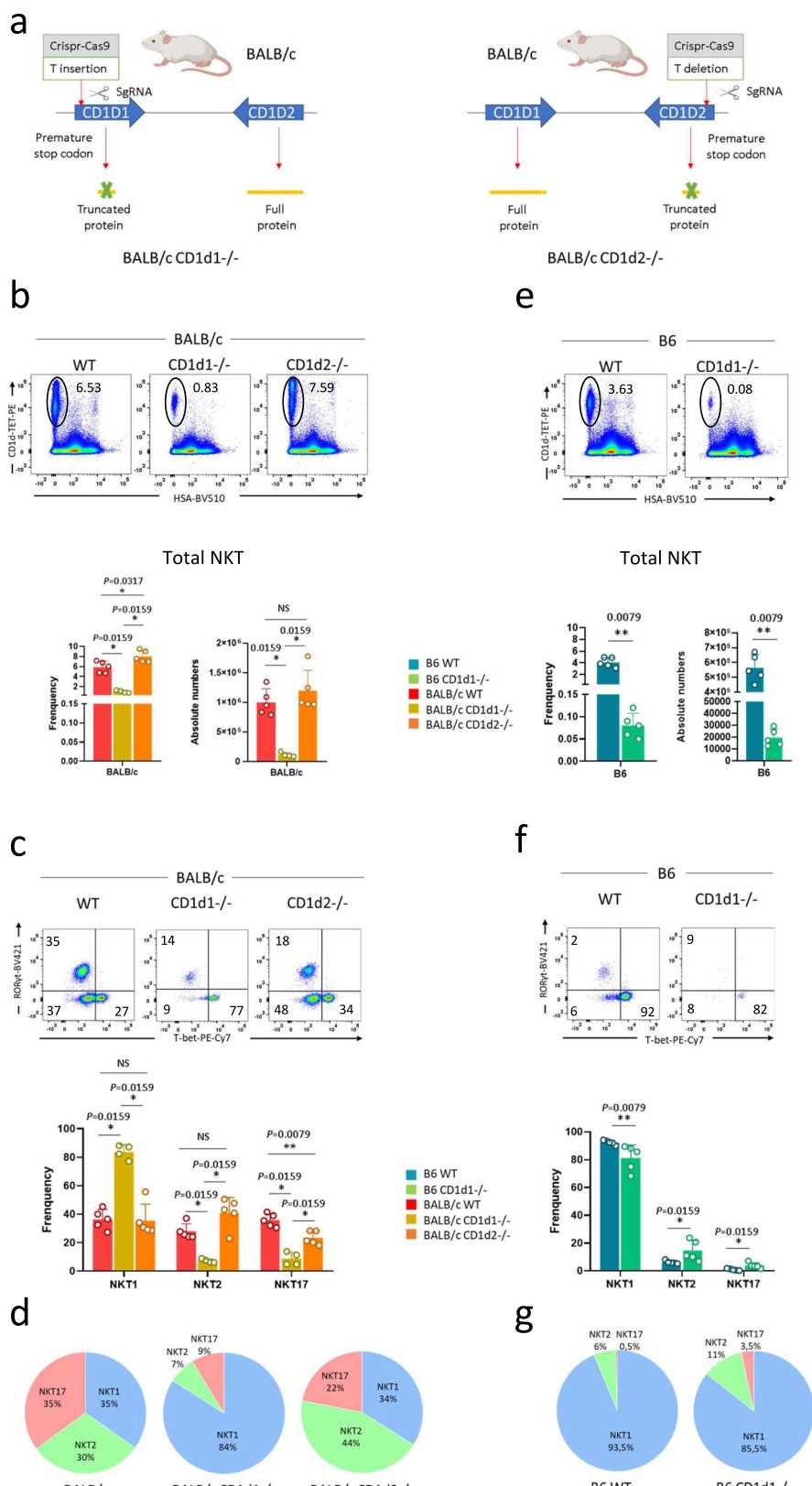

**Fig. 2 | CD1d1 and CD1d2 influence on NKT cell selection and distribution.**
**a** Schematic of *Cd1d1* and *Cd1d2* status in the mouse strains indicated.
**b**–**d** Representative staining of total NKT cells (CD1d-tet+HSA-) **b** or of NKT1 (T-bet[+]), NKT2 (T-bet[-]RORγt[-]), and NKT17 (RORγt[+]) subsets among thymic NKT cells from BALB/c WT (*n* = 5), BALB/c CD1d−/− (*n* = 4), and BALB/c CD1d2−/− (*n* = 5) mice **c**. Values indicated on dot plots represent frequencies. Individual/mean + SEM of mean frequency and absolute numbers for total NKT cells **b**, and individual/

mean + SEM of mean frequency for NKT cell subsets **c** are shown below. The latter results are also shown in pie chart for clarity **d**. **e**–**g** As in b-d for B6 WT (n = 5) and CD1d−/− (n = 5) mice. Data are representative of four experiments in b-d, and three experiments in e-g with 7-8-week-old mice. Statistics were calculated with the nonparametric Mann-Whitney test, two-sided, *P < 0.05, **P < 0.01. NS not significant (P > 0.05). Source data are provided as a Source Data file.

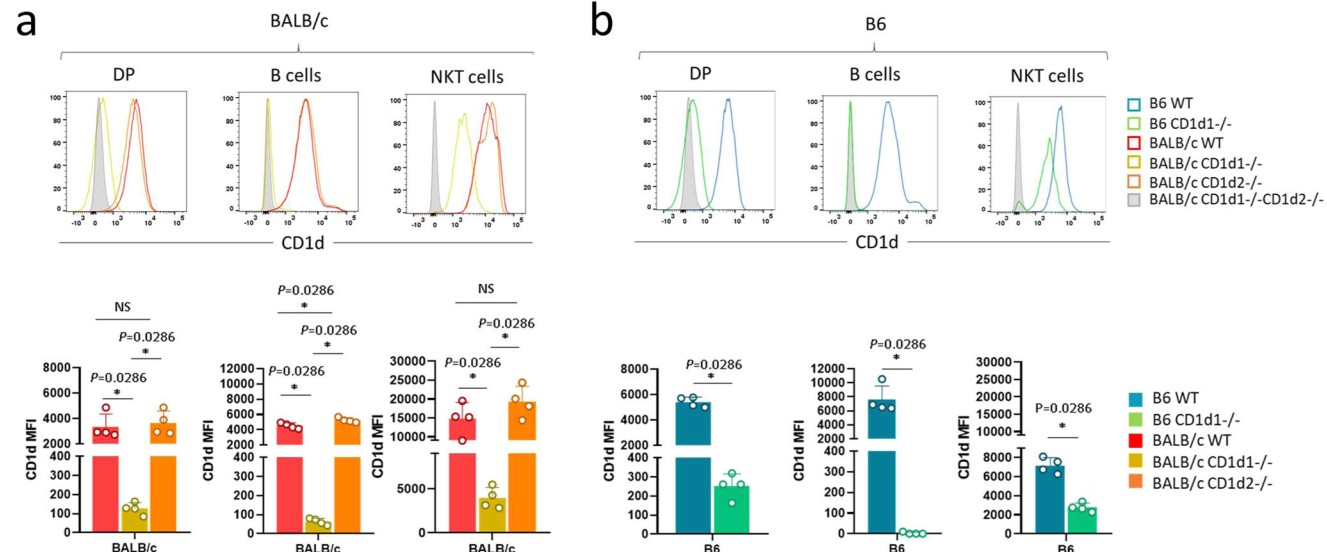

**Fig. 3 | CD1d expression in CD1d1 and CD1d2 deficient mice. a–b** Representative staining for CD1d expression on cortical thymocytes (DP), B cells, and NKT cells from BALB/c WT (n = 4), BALB/c CD1d1−/− (n = 4), and BALB/c CD1d2−/− (n = 4) **a** or B6 WT (n = 4), and B6 CD1d1−/− (n = 4) **b** mice. Individual/mean + SEM of mean MFI for these markers are shown below. Data are representative of three experiments with 7-8-week-old mice. Statistics were calculated with the nonparametric Mann-Whitney test, two-sided, *P < 0.05. NS not significant (P > 0.05). Source data are provided as a Source Data file.

reference BALB/c pattern when analyzing BM-reconstituted BALB/c mice. Overall, these results indicate that factors intrinsic to hematopoietic cells contribute to the strain-specific NKT cell subset distribution profile.

### CD1d-mediated ligand recognition alone does not control strain-specific differentiation and distribution

To further distinguish between the roles played by factors intrinsic and extrinsic to NKT cells in controlling their differentiation in B6 and BALB/c BM cells, we used mixed chimeras. To produce these, we infused B6 and BALB/c BM cells mixed at a 1:1 ratio to chimerically reconstitute lethally-irradiated F1 (B6 × BALB/c) CD1d1/2−/− mice (see scheme in Fig. 5a). The distribution of thymic and splenic NKT cells derived from B6 BM or BALB/c BM resembled the distribution observed in the respective single chimeric mice shown in Fig. 4a (Fig. 5b, dot plots and histograms). Thus, NKT cell differentiation and distribution appears to be intrinsic to BM-derived NKT cells.

To directly test the role of CD1d-ligand recognition in NKT cell differentiation, we generated mixed BM chimeras by admixing BALB/c CD1d1−/−CD1d2−/− (H-2Kd) BM cells or B6 CD1d1−/−CD1d2−/− (H-2Kb) BM cells, respectively, with WT B6 or BALB/c BM cells. These mixtures were then transferred into lethally-irradiated F1: (B6 x BALB/c) CD1d1/2−/− recipients (see scheme in Fig. 5a). CD1d-deficient DP thymocytes are known to lack the ability to select NKT cells, but they can generate NKT cells in the presence of CD1d-sufficient B6 or BALB/c DP thymocytes. We analyzed the thymic and splenic distribution of NKT1 and NKT2 cells developed from the CD1d1−/−CD1d2−/− BM compartment, on either BALB/c or B6 background (Fig. 5b, dot plots and histograms). The results showed a similar NKT cell subset distribution in thymus and spleen to that observed in the respective reference control BALB/c BM or B6 BM single chimeric mice (Fig. 4a). These results indicate that TCR/CD1d-ligand recognition does not direct strain-specific differentiation and distribution of NKT cell subsets.

### Multiple NKT cell precursor intrinsic factors are likely to be involved in differential strain-related subset distribution

Overall, the results presented indicate that CD1d-ligand recognition alone is not sufficient to promote strain-specific differentiation and subset distribution, and that additional factors intrinsic to the NKT cells themselves contribute to this process. We analyzed intrinsic factors related to the development and/or function of NKT1 (T-bet, IL-15R), NKT2 (IL-17RB: IL-25R), and NKT17 (RORγt, IL-7R, TGFbRII, phospho-SMAD2/3). We found no differences in expression levels that could explain the differential distribution of NKT cell in B6 vs BALB/c mice (Supplementary Fig. 3). However, our analysis revealed GATA3 expression levels to be around two-fold higher in NKT2 cells from BALB/c mice compared to B6 mice (Fig. 6a). Conditional deficiency of GATA3 is reported to result in cell-intrinsic defects in the thymic development of NKT cells[33]. Importantly, these mice lack stage 2 NKT cells, and GATA3 has been shown to be required for the development of CD4 NKT cells. Interestingly, stage 2 NKT cells comprise NKT2 cells, and NKT2 cells are mainly CD4+. In this light, the results found here strongly suggest that GATA3 could play a specific role in the development of NKT2 cells. It is therefore tempting to speculate that the higher expression of GATA3 in BALB/c mice could contribute in part to the prevalence of NKT2 cells in these mice. To confirm this hypothesis, it will be important to re-examine the distribution of NKT cell subsets in GATA3-deficient mice, using transcription factor expression to discriminate between NKT cell subsets in future studies.

Another factor that was shown to favor NKT2 cell development, reported by the Latour and Leite-de-Moraes groups[34], is SLAM-associated proteins (SAP). By taking advantage of SAP-deficient mice expressing a Vα14-Jα18 TCRα transgene, these authors found that SAP is critical not only for IL-4 production but also for the terminal differentiation of IL-4-producing NKT2 cells[34]. Based on their findings, the authors propose that SAP-dependent signals are essential for the fate decisions driving NKT2 cell differentiation. Here, we detected higher expression of SLAM proteins at the cell surface of NKT2 cells from BALB/c compared to B6 mice (Fig.1c and Supplementary Fig. 1d). This expression pattern could in part contribute to the predominance of NKT2 cells in BALB/c mice.

The Enhancer of zeste homolog 2 (EZH2) is another factor that has been shown to control NKT2 cell development[35]. EZH2 is a member of the polycomb repressive complex 2 and is enzymatically responsible for methylating lysine 27 on histone 3 (H3K27me3), a key epigenetic modification known to repress lineage-determining transcription factors during cellular differentiation[36,37]. Loss of EZH2 and H3K27me3 in NKT cells resulted in increased NKT cell numbers and preferential

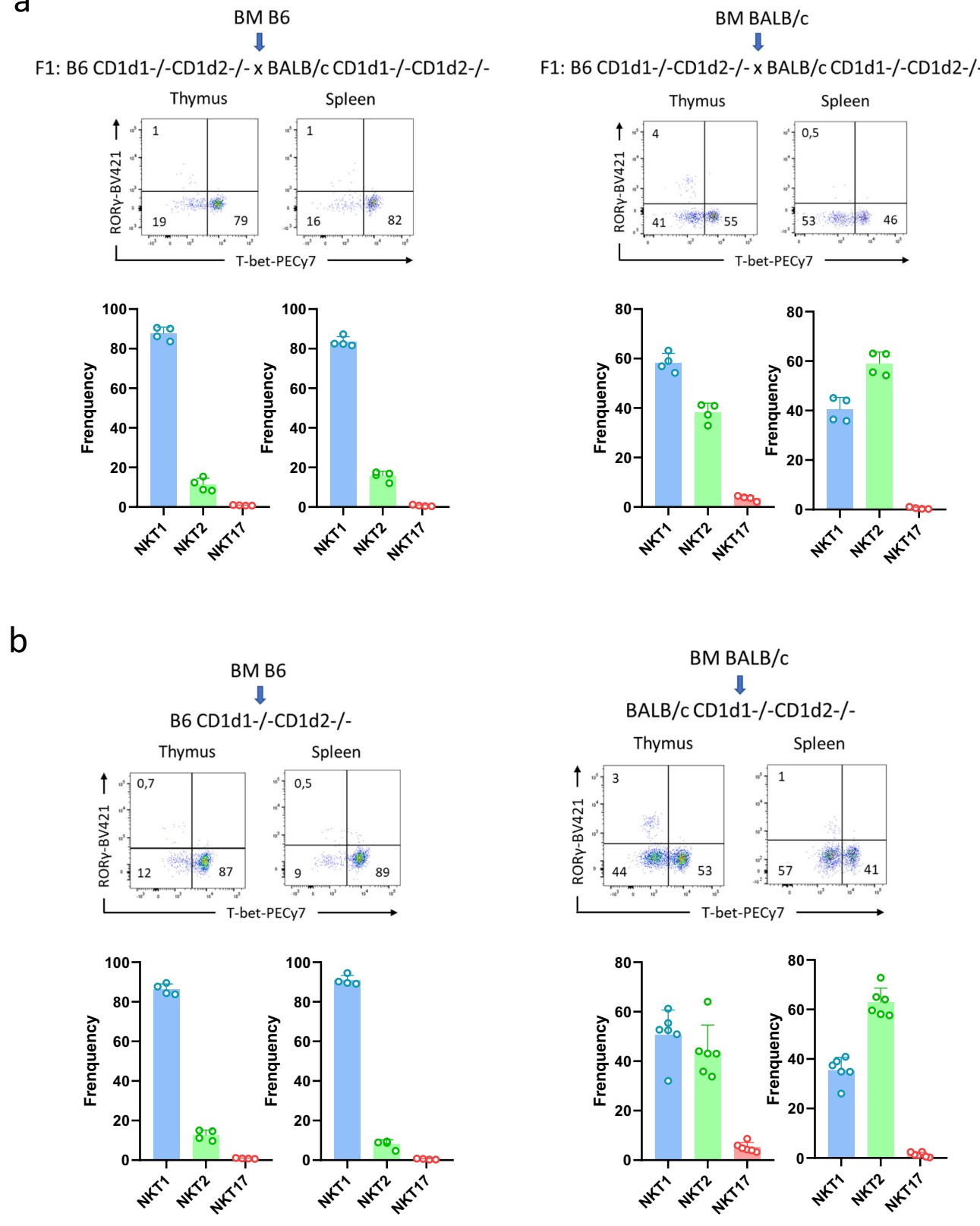

**Fig. 4 | Factors intrinsic to hematopoietic cells contribute to the differential NKT cell subset distribution profile. a** Representative staining of NKT1 (T-bet[+]), NKT2 (T-bet[-]RORγt[-]), and NKT17 (RORγt[+]) subsets among thymic and splenic NKT cells from the indicated left (*n* = 4) and right (*n* = 4) bone marrow chimeras. Numbers on dot plots represent frequencies. Individual/mean + SEM of mean frequency for each thymic and splenic NKT cell subset are shown below. **b** Representative thymic and splenic staining as in **a**. with *n* = 4 and *n* = 6 for the indicated left and right chimeras, respectively. Data are representative of three experiments in **a**. and five experiments in **b**. with 7-8-week-old mice. Source data are provided as a Source Data file.

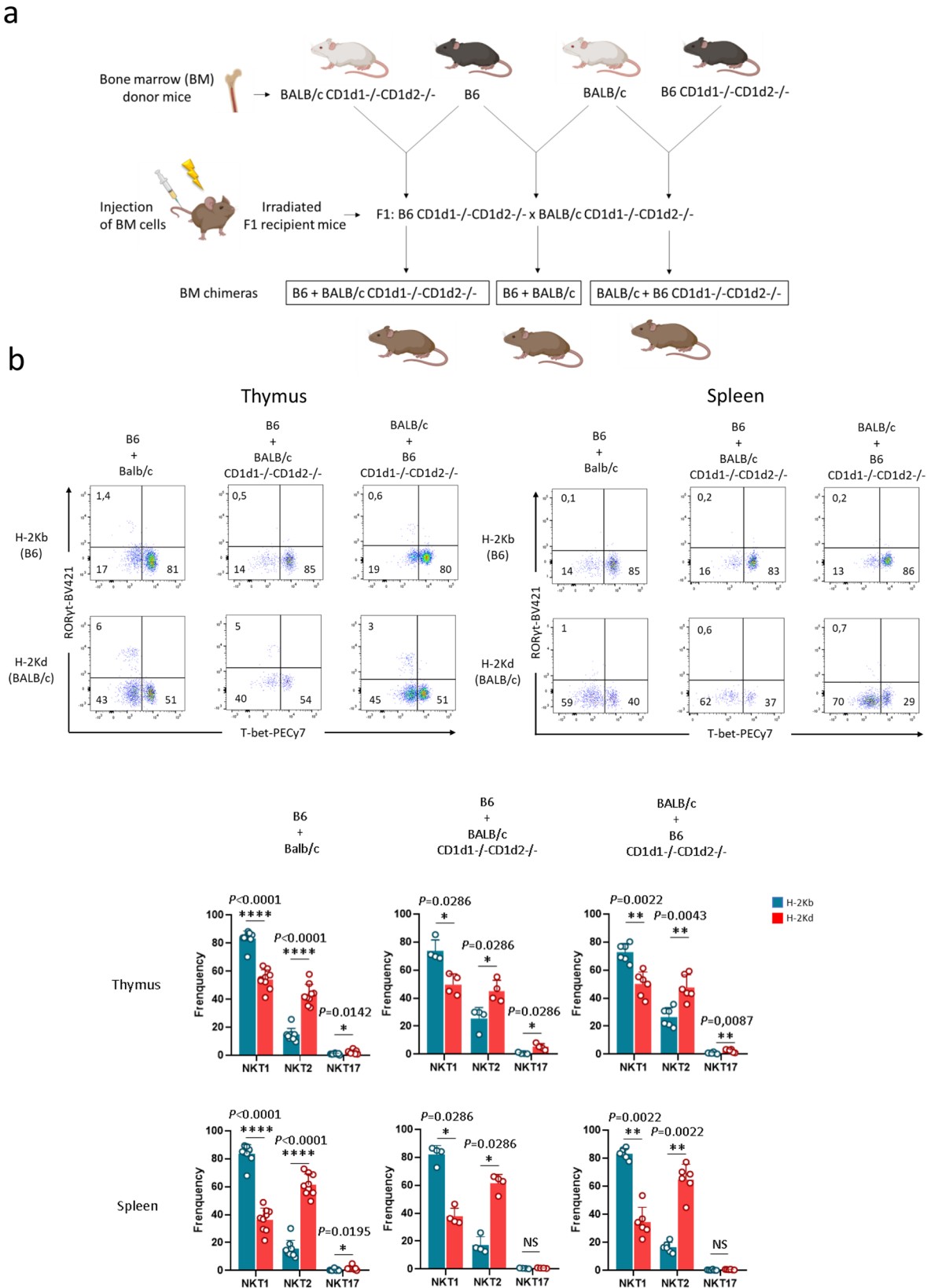

development into NKT2 cells[35]. Using a published data set produced by Georgiev et al.[38], analyzing transcriptional expression of NKT cell subsets in B6 vs BALB/c mice, we found that EZH2 transcript expression is higher in B6 NKT2 cells compared to BALB/c NKT cells (Fig. 6b). This result was corroborated by intracellular staining analysis—which showed a higher H3K27me3 level in B6 NKT2 cells compared to BALB/c

NKT cells (Fig. 6c). The reduced protein levels in BALB/c NKT cells could contribute in part to the higher NKT2 cell frequency in BALB/c mice.

Exploiting the Georgiev et al.[38] data set, we also found NRP2 transcripts (encoding for neuropilin-2) to be highly expressed in BALB/c NKT2 cells compared to B6 mice (Fig. 6d)[39]. It was impossible to

**Fig. 5 | CD1d-mediated ligand recognition does not influence NKT cell specification. a** Schematic of chimeric mice generation. **b** Representative staining for NKT1 (T-bet+), NKT2 (T-bet−RORγt−), and NKT17 (RORγt+) subsets among thymic and splenic NKT cells in B6 (H-2Kb) and BALB/c (H-2Kd) compartments from the B6 + BALB/c (n = 9), B6 + BALB/c CD1d1−/−CD1d2−/− (n = 4), and BALB/c + B6 CD1d1−/−CD1d2−/− (n = 6), mixed bone marrow chimeras. Individual/ mean + SEM of mean MFI for these markers are shown below. Statistics were calculated with the nonparametric Mann-Whitney test, two-sided, frequency thymus NKT1, NKT2 and NKT17 H-2Kb (B6) vs. H2-Kd (BALB/c) in B6 + BALB/c chimeras: ****P < 0.0001, ****P < 0.0001, *P = 0.0142; frequency thymus NKT1, NKT2 and NKT17 H-2Kb (B6) vs. H2-Kd (BALB/c) in B6 + BALB/c CD1d1−/− −CD1d2−/− chimeras: *P = 0.0286 for all subsets; frequency thymus NKT1, NKT2

and NKT17 H-2Kb (B6) vs. H2-Kd (BALB/c) in BALB/c + B6 CD1d1−/−CD1d2−/− chimeras: **P = 0.0022, **P = 0.0043, **P = 0.0087. frequency spleen NKT1, NKT2 and NKT17 H-2Kb (B6) vs. H2-Kd (BALB/c) in B6 + BALB/c chimeras: ****P < 0.0001, ****P < 0.0001, *P = 0.0195; frequency spleen NKT1, NKT2 and NKT17 H-2Kb (B6) vs. H2-Kd (BALB/c) in B6 + BALB/c CD1d1−/−CD1d2−/− chimeras: *P = 0.0286, *P = 0.0286, NS: not significant (P > 0.05); frequency spleen NKT1, NKT2 and NKT17 H-2Kb (B6) vs. H2-Kd (BALB/c) in BALB/c + B6 CD1d1−/ −CD1d2−/− chimeras: **P = 0.0022, **P = 0.0022, NS: not significant (P > 0.05). Data are representative of four experiments (B6 + BALB/c chimeras), and five experiments (B6 + BALB/c CD1d1−/−CD1d2−/− and BALB/c + B6 CD1d1−/ −CD1d2−/− chimeras) with 7-8-week-old mice. Source data are provided as a Source Data file.

directly confirm NRP2 expression by FACS due to a lack of appropriate antibody. However, we exploited the fact that NKT2 cells constitutively produce IL-4 to confirm high expression of NRP2 in BALB/c NKT2 cells compared to B6 mice by immunohistochemistry. For these experiments, we used an IL-4/GFP-enhanced transcript (4Get) strain, expressing an IL-4-triggered fluorescent reporter, to visualize NKT2 cells (Fig. 6e). Although neuropilins and semaphorins are mainly known as modulators of axon guidance, angiogenesis, and organogenesis in the developing nervous system, they also play a role in the immune system[40]. NRP2 mainly binds to semaphorin 3F and 3C (Sema3F, 3C)[40]. In humans, NRP2/Sema3F axis was reported to inhibit thymocyte migration in response to S1P1, a chemokine with a well-documented role in thymocyte egress from the thymus[39]. Because BALB/c NKT2 cells express NRP2, this may render emigration of NKT2 inefficient in this strain, potentially explaining why NKT2 cells are more frequent in BALB/c mice. Future studies using NRP2-deficient mice will allow the functional consequences of NRP2 expression in BALB/c NKT2 cells to be verified.

### CD1d expression levels influence NKT cell subset composition and function

To assess the influence of CD1d expression on NKT cell differentiation, we analyzed NKT cell development in CD1d heterozygous mice (CD1d +/−), where CD1d density is reduced by half compared to WT mice. We performed this study in F1: B6 CD1d1−/−CD1d2−/− x BALB/c mice, which presented two advantages. First, the F1 mice (CD1d+/−) have sizable populations of NKT2 and NKT17 cells compared to B6 mice; secondly, it avoids the need to perform separate studies in B6 and BALB/c backgrounds. Controls were CD1d+/+ F1 mice produced by crossing B6 and BALB/c mice; both F1 animals therefore shared a B6/ BALB/c mixed background (Fig. 7a).

In CD1d+/− mice, we observed a normal thymic T cell distribution based on CD4 and CD8 expression (Fig. 7b, upper panel). As expected, CD4 + CD8 + DP cortical thymocytes expressed CD1d molecules at an intensity corresponding to a 2-fold reduced level (Fig. 7b, lower panel). Compared to WT controls, NKT cells were present in these CD1d+/− F1 mice at the same frequency and absolute numbers (Fig. 7c), indicating that normal NKT cell selection occurs in these mice. However, we found an altered distribution of NKT cells in these mice, with an increase in the frequency of NKT2 cells associated with a decreased frequency of NKT17 cells. The frequency of NKT1 cells remained unchanged (Fig. 7c).

As expected, CD1d expression affected TCR signal strength in CD1d+/− F1 animals, as reflected by the levels of CD5, PLZF and Egr2, which were reduced compared to control mice in total NKT (Fig. 7d), and in each of the NKT cell subsets (Supplementary Fig. 4). The α chain in NKT cells is constant, but analysis of the TCR β chain repertoire in the total NKT cells and NKT cell subsets revealed an increased frequency of Vβ7 in total NKT cells from CD1d+/− compared to CD1d+/+ controls (Fig. 7e, representative dot plots and histograms). In CD1d+/+ NKT cell subsets, analysis of Vβ chain usage showed the following hierarchy of Vβ7 usage NKT2 > NKT1 > NKT17. This ranking is

consistent with results from a previous study[6]. In CD1d+/− mice compared to CD1d+/+ mice, Vβ7 use was increased in NKT2 and NKT1, but not in NKT17 cells (Fig. 7e, representative dot plots and histograms). In contrast, we observed no changes in Vβ8 usage. (Fig. 7f, representative dot plots and histograms). Our results show that Vβ7 use correlated with differences in NKT2 and NKT17 frequency, but not NKT1 cell frequency in CD1d+/− mice. Indeed, the subsets skewed toward Vβ7 expression had unaltered (NKT1) or increased (NKT2) frequencies, whereas the subset with the lowest Vβ7 expression (NKT17) was present at a reduced cell frequency.

To assess the functional capabilities of the NKT cells present in CD1d+/− mice, we analyzed cytokine production in these subsets after stimulation with PMA/Ionomycin. NKT cells expressing NK1.1, representing mature NKT1 cells (NK1.1+), had an unaltered IFN-γ−production potential (Fig. 8a, upper panel). The NK1.1-CD138-subset – comprising T-bet+ immature NKT1 cells and NKT2 cells—also had the same potential to produce IFN-γ as control mice (this cytokine is produced by immature T-bet+ NKT1 cells) (Fig. 8a, middle panel). In contrast, they had a reduced capability to produce IL-4 (produced only by NKT2 cells as the aforementioned mature NK1.1 + NKT1 cells do not produce IL-4) (Fig. 8a, middle panel). As for CD138 + NKT17 cells, these cells had a reduced capacity to produce IL-17 (Fig. 8a, lower panel). To assess the consequences of reduced functional capabilities of NKT cells in CD1d+/− mice, we analyzed the development of Eomes-positive CD8 virtual memory cells. Development of this population largely depends on IL-4 produced by NKT2 cells[6]. We found the reduced expression of IL-4 in CD1d+/− mice to correlate with a decreased frequency and absolute number of CD8+Eomes+ cells in the thymus (Fig. 8b).

Overall, these results show that CD1d-expression levels could not only influence NKT cell subset composition, but also the acquisition of their effector function.

## Discussion

In this study, we aimed to determine whether CD1d-ligand recognition by developing NKT cells could influence their NKT1, 2, or 17 differentiation profile, and could thus explain the different NKT cell subset distribution between B6 (where NKT1 predominate) and BALB/c (where NKT2/NKT17 predominate) mouse strains. Our results showed that this differential subset distribution is not linked to the expression of CD1d2 molecules—present in BALB/c and not B6 mice. Rather, we found the differential distribution to be driven by intrinsic NKT cell factors and by CD1d-ligand expression levels linked to NKT cell TCR affinity.

TCR strength/duration regulates cell fate decision during T cell development in the thymus, leading to the development of helper CD4 and cytotoxic CD8 T cells[41]. TCR strength is also thought to influence cell fate decisions for functional γδ T cells producing IL-17 and IFN-γ[42,43], as well as NKT cell subset differentiation[17,23]. Reduced TCR strength due to altered components of the signaling cascade downstream of the TCR in naturally mutant or genetically modified mice differentially affected NKT cell subset development[17,23]. In these mice, NKT2 but not NKT1 cell development is altered, suggesting that a

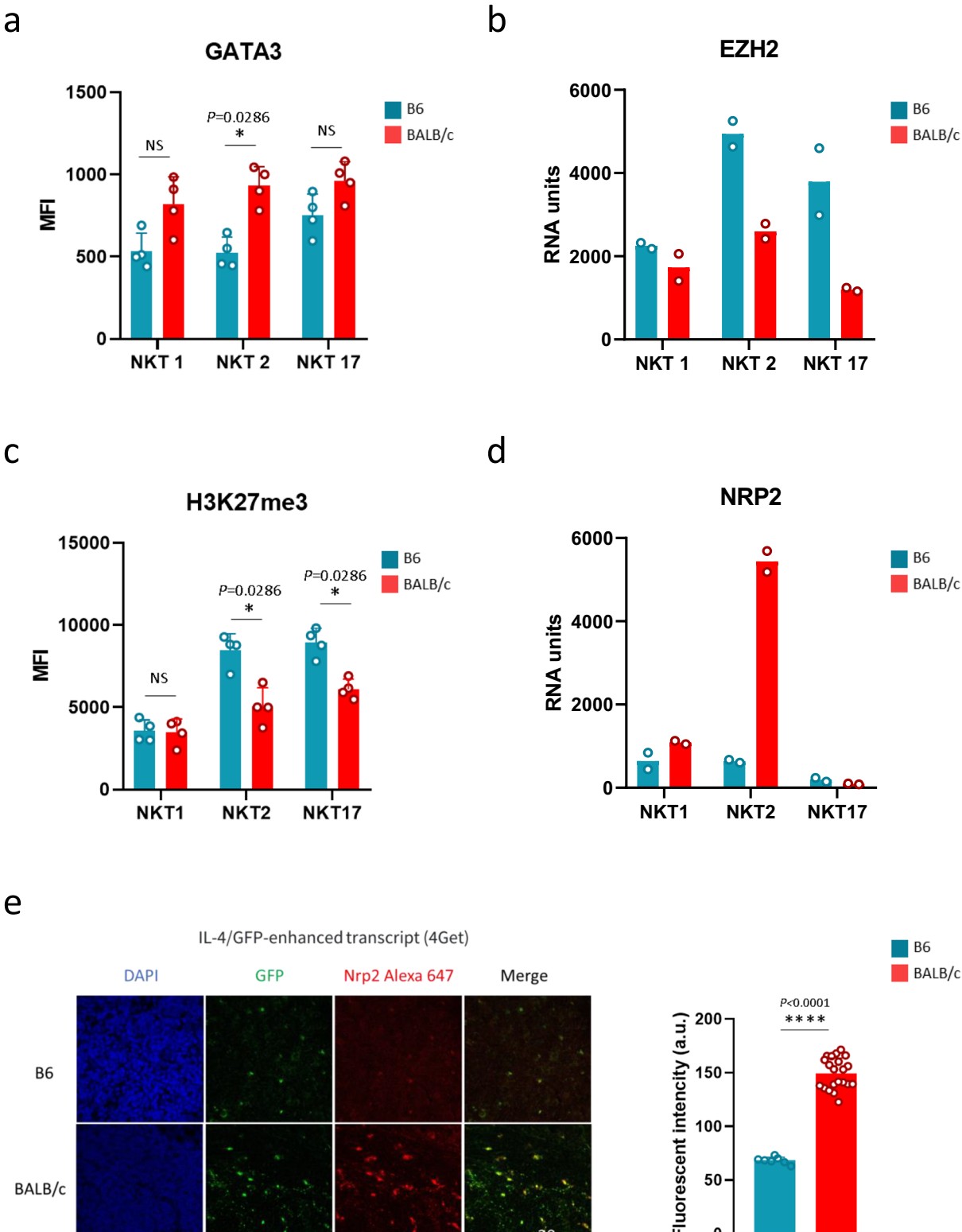

strong/sustained TCR signal is required for NKT2 cell development. These results fit with reports that NKT2 and NKT1 cells perceive the highest and lowest TCR signal strength in normal mice, respectively[17,23]. Interestingly, our results showed a higher TCR signal strength perceived by cells in NKT2-rich BALB/c than NKT1-rich B6 mice, suggesting that TCR signal strength plays a role in NKT cell fate decision, and providing a potential link with strain-specific NKT cell subset distributions.

Several studies indicate that the TCR signal perceived by NKT cells is potentiated by homotypic SLAM/SLAM interactions[15–17]. We found SLAM expression to be higher on BALB/c compared to B6 NKT cells (see Fig. 1c). This could potentially explain the higher TCR signal strength perceived by BALB/c NKT cells. However, this possible explanation does not fit with the results of another study showing that SFR dampen TCR signaling during NKT cell development[18,19]. We also found CD1d expression levels in BALB/c DP cells to be reduced

**Fig. 6 | Differential expression levels of key intrinsic factors involved in NKT cell subset development and/or function in B6 vs BALB/c mice. a** GATA3 expression. Individual/mean + SEM of mean MFI for GATA3 in the indicated NKT cell subsets and mouse strain. **b** EZH2 mRNA expression profiles in the indicated subsets and mouse strains. Chip-based transcriptome analysis was performed as described in Georgiev et al. (ref. [38]). RU (RNA units) represents an arbitrary indicator of expression strength. Data are from two independently-performed transcriptome analyses for each strain: BALB/c and B6 (*n* = 2). **c** H3K27me3 expression. Individual/mean + SEM of mean MFI for H3K27me3 in the indicated NKT cell subsets and mouse strains. **d** NRP2 mRNA expression profiles. Same as in **b**. Statistics were calculated with the nonparametric Mann-Whitney test, two-sided, \**P* < 0.05.

NS not significant (*P* > 0.05). **e** Thymic sections from B6 (top) and BALB/c (bottom) 4get mice, showing detection of NKT2 cells (GFP+), stained with anti-Nrp2 antibody and counterstained with DAPI. Scale bar: 20 μm. The histogram shows the mean fluorescence intensity level for Nrp2 above background. *n* = 21 and 7 cells for BALB/c and B6 tissues, respectively. Data in **a** and **c** are representative of three experiments on 7-8-week-old mice, *n* = 4 per strain in each experiment. Data in **e** are representative of two independent experiments. Statistics were calculated with the nonparametric Mann-Whitney test, two-sided, fluorescence intensity (a.u.) B6 vs. BALB/c: \*\*\*\**P* < 0.0001. NS not significant (*P* > 0.05). Source data are provided as a Source Data file.

compared to B6 DP cells (see Fig. 1d). This profile could not explain the high TCR signal strength perceived by BALB/c NKT cells.

It is possible that the CD1d2 isomorph, expressed only in BALB/c mice, provides a differential TCR signal, and most significantly influences NKT2/NKT17 differentiation in BALB/c mice. To test this hypothesis, we generated BALB/c CD1d1−/− mice (expressing CD1d2). In these mice, NKT cell frequencies and numbers were reduced, correlating with the reduced CD1d2 expression compared to BALB/c CD1d2−/− mice (expressing CD1d1). The latter animals had comparable CD1d expression and NKT cell frequencies to wild type BALB/c mice.

We do not know the exact reasons for the reduced CD1d2 expression in CD1d1−/− mice. Intracellular staining showed no accumulation of CD1d2 molecules in DP thymocytes from BALB/c CD1d1−/− mice (Supplementary Fig. 5a). Transcriptional analysis showed equal amounts of CD1d2 and CD1d1 transcripts in BALB/c CD1d1−/− DP thymocytes, as observed in previous studies of BALB/c DP thymocytes (Supplementary Fig. 5b, pie chart)[29]. In addition, equal amounts of CD1d2 transcripts were detected in BALB/c CD1d1−/− mice and BALB/c mice, hinting at a translational rather than a transcriptional issue behind the low expression of CD1d2 in the thymus (Supplementary Fig. 5b, histograms). This notion is supported by the finding that CD1d2 expression is upregulated in selected thymic NKT cells compared to the DP thymocytes from which they originate (Fig. 3a, right histograms and histogram plots). This upregulation also applies to NKT cells developed in B6 CD1d1−/− (expressing CD1d2) (Fig. 3b, right histograms and histogram plots), and correlates with positive selection, as CD1d is also upregulated in NKT cells compared to DP cells in normal wild type B6 or BALB/c mice. This phenomenon is not restricted to NKT cells or related to their agonist selection. Indeed, it is reminiscent of MHC class I expression, which is undetectable in DP thymocytes and upregulated in selected CD4 or CD8 T cells[44].

To understand the importance of CD1d-antigen recognition in determining NKT lineage, we generated B6 and BALB/c mixed chimeras and mixed chimeras where either one of the compartments was CD1d-deficient. The results obtained showed that NKT cell fate is dictated by factors intrinsic to the NKT cells themselves, and that allogenic CD1d-recognition cannot reverse this developmental fate. Surprisingly, we found that, whereas BM B6 into B6 chimeras produced a similar NKT cell distribution in the thymus and spleen to WT B6, BALB/c BM into BALB/c chimeras showed a shift in thymic populations toward a predominance of NKT1 lineage, and a low NKT17 frequency. In contrast, a high frequency of NKT2 cells was maintained in the chimera. These results are consistent with a previous study showing the same non-classical thymic NKT cells distribution in syngeneic BALB/c BM chimeric mice (increased NKT1 and reduced NKT17 cells)[17]. We do not know the reasons for the altered distribution. One explanation could be that NKT17 cells have an embryonic window of development, and that they do not develop in adult mice. NKT1 cells could therefore expand to occupy the space left by NKT17 cells. However, this is likely not the case, as NKT17 cells developed in BM B6 into B6 Jα18−/− chimeric mice (Supplementary Fig. 6). In addition, although in the spleen of these BALB/c BM into BALB/c chimeras we observed a more classical BALB/c-like NKT cell subset distribution

pattern, with a dominance of NKT2 over the NKT1 subset, the NKT17 cell frequency remained low compared to that observed in the spleen of WT BALB/c mice[45]. Future studies will be needed to solve this issue.

Our results thus indicate that multiple intrinsic factors are likely to be involved in differential strain-related subset distribution - including transcription factors, down-stream signaling cascades, epigenetic factors and migratory factors, among others. The fine-tuning of these intrinsic factors can be linked to gene polymorphisms and/or regulatory enhancer/silencer regions. To uncover new genes controlling NKT cell differentiation and strain distribution, we will deploy an unbiased genetic approach, using a collection of mice generated by a consortium of mouse geneticists and laboratories[46]. This collection (the CC collection) of inbred strains was produced by crossing eight founder strains, three of which were established from different mouse subspecies. Using this resource, we will have access to a genetic diversity that will allow us to correlate NKT cell subset distribution and genetic controls. This collection of mouse strains has previously been successfully used to study the genetic control behind NK cell phenotype and function variability observed in different strains[47].

To determine a role for CD1-ligand recognition in NKT cell development, we analyzed NKT cell selection and differentiation in CD1d +/- mice expressing 50% lower surface levels of CD1d on a mixed B6 CD1d1−/−CD1d2−/− x BALB/c background. In this model, there was no interference with genetic background. The frequencies and numbers of NKT cells remained unchanged in these mice, in agreement with results obtained in wild type NOD and NOD.CD1d+/− mice[48], where no significant differences between NKT cell frequencies were reported between the two strains. However, in this previous study, there existed a negative relationship between CD1d expression levels and the frequency of thymic NKT cells in different mouse strains[48]. Here, the B6 and BALB/c mice analyzed appeared to fall within this linear relationship, and results confirmed that physiological variations in CD1d expression play an important role in controlling NKT cell development. However, simply reducing its expression is not sufficient to increase the frequency of these cells.

Nevertheless, we found that reduced CD1d levels did affect NKT cell subset distribution, favoring cells expressing a Vβ7-containing TCR. Thus, NKT2 cells were more frequent than NKT17 subsets. Our results are in agreement with previous studies showing that Vβ7-containing NKT TCR have a higher affinity for ligands, and confirm the importance of the Vβ chain in NKT cell subset composition[49,50]. It is possible to consider that lower CD1d expression levels in BALB/c DP thymocytes compared to B6 mice could contribute in part to the predominance of the NKT2 cell subset in these mice, but this reasoning does not apply to NKT17 cells. Thus, physiological variations in CD1d expression levels could influence NKT cell distribution but cannot account for differences in subset distribution in distinct mouse strains, highlighting the influence of other genetic variants. Data from other laboratories suggest that decreased TCR signal during selection will promote NKT1 cells and conversely disfavor NKT2 cells[17,23]. CD1d+/− mice, which have a decreased signal, do not however have fewer NKT2 cells. This is likely due to the fact that the defect in TCR signal in these mice is extrinsic to NKT cells (reduced CD1d expression in selecting DP

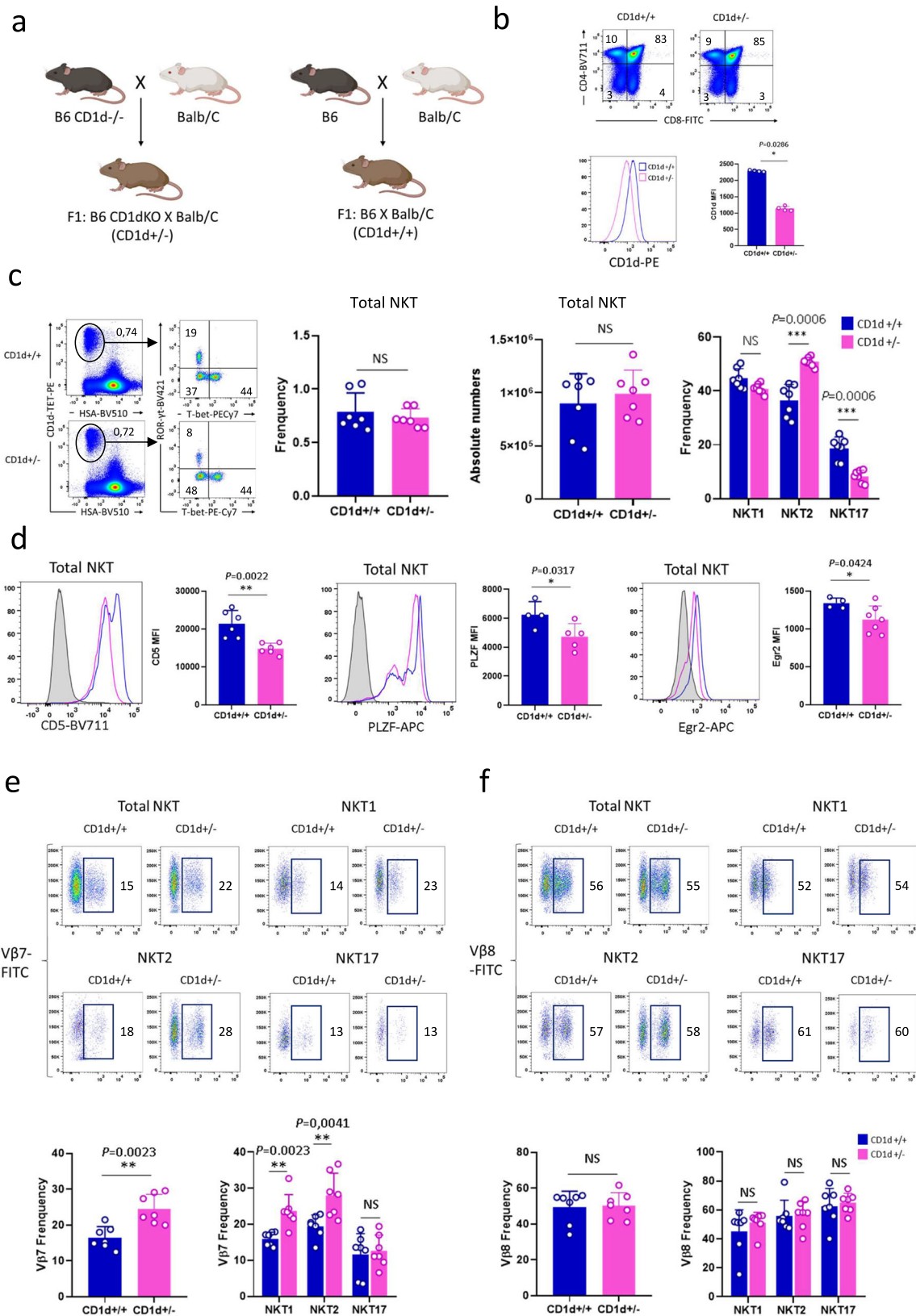

cells) contrary to previous studies where mouse models expressing hypomorphic alleles of Zap70 were studied[17,23]. In CD1+/−mice, where components of TCR signaling cascade are not altered, NKT2 cells likely adapt to the reduced CD1d expression by increased usage of Vβ7 that confer to these cells a higher affinity for ligand and allows their increased frequency. It is interesting to note that NKT2 cells in CD1d+/−

mice still perceive the highest signal among NKT cells, indicating that TCR signal strength hierarchy among NKT cell subsets is maintained.

Despite this signal strength, analysis of CD1d+/− mice showed that NKT2 and NKT17 cells, which perceive a strong TCR signal, were functionally affected, and that both produced less cytokine. In contrast, NKT1 cells, which perceive a weaker TCR signal, produced IFN-γ

**Fig. 7 | CD1d expression levels affect NKT cell subset composition. a** Crossing strategy used to obtain the indicated mice. **b** Upper panel: representative staining for CD4 and CD8 expression in thymocytes from CD1+/+ ($n = 4$) and CD1d+/− ($n = 4$) mice. Numbers on dot plots correspond to frequencies. Lower panel: Representative staining for CD1d expression on DP cortical thymocytes from CD1+/+ ($n = 4$) and CD1d+/− ($n = 4$) mice. Individual/mean + SEM of mean MFI are shown on the right. **c** Representative staining for NKT1 (T-bet⁺), NKT2 (T-bet⁻RORγt⁻), and NKT17 (RORγt⁺) subsets among thymic NKT cells from CD1d+/+ ($n = 7$) and CD1d+/− ($n = 7$) mice. Numbers on dot plots represent frequencies. Individual/mean + SEM of mean frequency and absolute numbers of total NKT cells and frequency in each NKT cell subset are shown in the right panel. **d** Representative staining for CD5, PLZF, and

Egr2 in total thymic NKT cells from CD1d+/+ ($n = 6$, 4, and 4 for CD5, PLZF, and Egr2, respectively) and CD1d+/− ($n = 6$, 5, and 7 for CD5, PLZF, and Egr2, respectively) mice. Individual/mean + SEM of mean MFI for these markers are shown in the right panel. **e, f** Representative staining for Vβ7 **e** and Vβ8 **f** expression on NKT cells and NKT cell subsets from CD1+/+ ($n = 7$) and CD1d+/− ($n = 7$) mice. Numbers on dot plots correspond to frequencies. Individual/mean + SEM of mean frequency of total NKT cells and of each NKT cell subset are shown below. Data are representative of four experiments with 7-8-week-old mice. Statistics were calculated with the non-parametric Mann-Whitney test, two-sided, *$P < 0.05$, **$P < 0.01$, ***$P < 0.001$. NS not significant ($P > 0.05$). Source data are provided as a Source Data file.

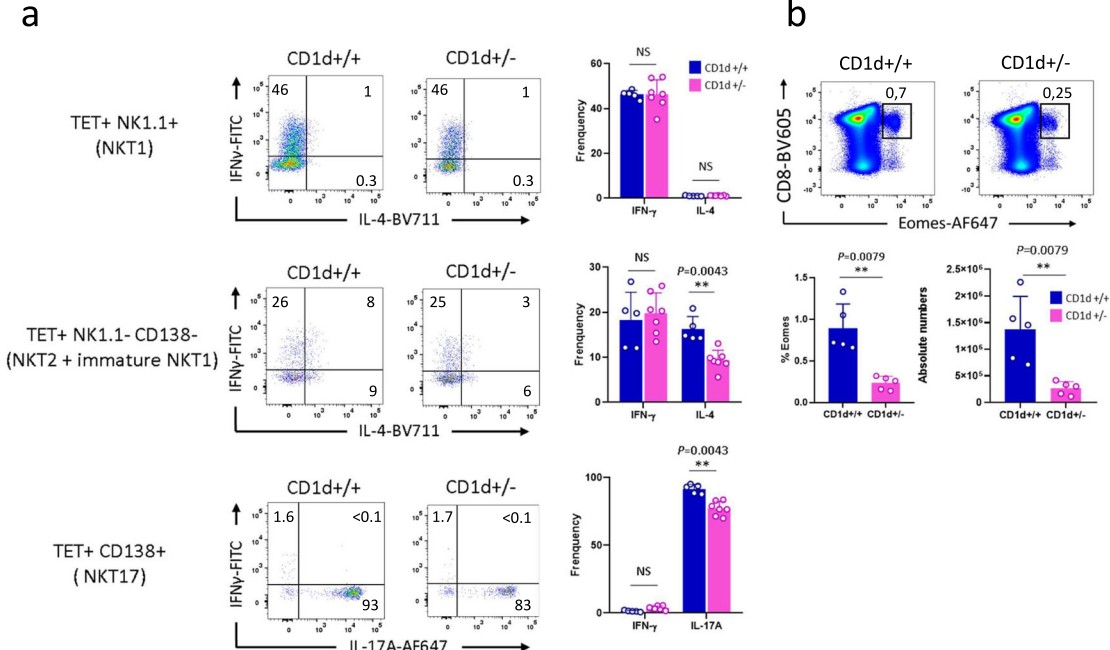

**Fig. 8 | CD1d-expression levels affect acquisition of NKT cell function.**
**a** Representative staining for IL-4, IFN-γ, and IL-17 among thymic NKT cell subsets from CD1d+/+ ($n = 5$ for NKT1, $n = 5$ for NKT2 + immature NKT1, and $n = 5$ for NKT17) and CD1d+/− ($n = 7$ for NKT1, $n = 7$ for NKT2 + immature NKT1, and $n = 7$ for NKT17) mice. Values indicated on dot plots represent frequencies. Individual/mean + SEM of mean frequency of cytokine-positive NKT cells are shown on the right.
**b** Representative Eomes staining pattern among thymic CD8-positive cells from

CD1d+/+ ($n = 5$) and CD1d+/− ($n = 5$) mice. Numbers on dot plots represent frequencies. Individual/mean + SEM of mean frequency and absolute numbers are shown below. Data are representative of three experiments on 7-8-week-old mice. Statistics were calculated with the nonparametric Mann-Whitney test, two-sided, *$P < 0.05$, **$P < 0.01$. NS not significant ($P > 0.05$). Source data are provided as a Source Data file.

at the same levels as in WT animals. These results reveal a correlation between TCR signal strength and functional acquisition of NKT cells subsets.

Overall, our results provide insights into the roles played by CD1d/TCR interactions in NKT cell differentiation. Although it was assumed that CD1d2 contributes to the development of NKT cells, our results suggest that this marker plays only a minor role in NKT cell development and subset distribution in BALB/c mice. In conjunction with CD1d/TCR interaction, we found that intrinsic NKT cell factors also influence strain-specific differentiation and subset distribution. Our findings suggest that various factors, such as migration characteristics and epigenetic modifications, may govern strain-associated differential distribution of NKT cell subsets. Future work will allow us to confirm the importance of the candidate factors identified, and to reveal new ones.

## Methods
### Mice
Jα18−/−, C57BL/6 CD1d1−/− and 4get mice are described elsewhere[51–53]. C57BL/6 and BALB/c CD1d1/CD1d2−/− mice were purchased from the

Jackson Laboratories. WT C57BL/6 and BALB/c mice were purchased from Janvier Laboratories; B6 Ly-5.1 mice were purch6ased from Charles River Laboratories. All mice were bred and maintained under specific pathogen free conditions, on a 12 h light/dark cycle at 20–24 degree and controlled humidity (30–70%, usually around 50%), and experiments were performed in accordance with the Institutional Animal Care and Use Guidelines. The study protocol was approved by the local ethics committee—Comite d'Ethique Paris-Nord (C2EA-121)—affiliated with the Comité National de réflexion Ethique en Expérimentation Animale and the French Ministry for higher education and research.

### Generation of *Cd1d1* and *Cd1d2* knock-out mice
BALB/c mice were purchased from Janvier Labs at 3 to 4 weeks of age to produce embryos. All experiments were performed in accordance with NIH guidelines, and European Union recommendations (2010/63/UE). Female BALB/c mice were injected with pregnant mare serum gonadotropin (PMSG) and human chorionic gonadotropin (hCG) with a 48-hour interval, before mating with male BALB/c mice. Cas9 mRNA (3 μM) and sgRNAs (200 ng/μL for each sgRNA) targeting exon1 of each

gene were mixed and electroporated (NEPA21) into the pronuclei of the one-cell embryos. For CD1d1, two guides were used: SgRNA1 5′-CCCACAGCAACAGCCATGGT-3′ and SgRNA3 5′-CCTACCATGGCTGTTGCTGT-3′. For CD1d2, the following two guides were used: SgRNA2 5′-CCTACCGTGCCTGTTGCTGT-3′ and SgRNA4 5′-CAGCAACAGGCACGGTAGGT-3′. After transfection, the zygotes were cultured in KSOM Medium at 37 °C–5% CO$_2$, until the blastocyst stage (around 100 cells) before transferring into pseudo-pregnant mice. This step was performed in the animal facilities at the Cochin Institute. When founders were identified, they were mated with BALB/c mice to obtain CD1D1-heterozygote animals. Founders were obtained with an insertion of a T base, causing a reading frame shift that created an early stop codon (residual 19-amino acid protein). For CD1D2, founders were obtained with a deletion of a T base causing a reading frame shift that produced an early stop codon (residual 38-amino acid protein). Mouse genotyping was verified by PCR and Sanger sequencing. For CD1D1, forward primer 5′-CCCTTCTCTAGATTGTGTGC-3′ and reverse primer 5′-CGGGAGCAGAGTAAAGCGCA -3′ were used. CD1D2 primers were: forward 5′-CCCTTCTCTAGATTGTGTGC-3′ and reverse 5′ CGGGAGCAGAGTAAAGCGCA -3.

## Cell preparation

Thymus, pooled peripheral lymph nodes (PLNs; comprising axillary, subaxillary, maxillary, inguinal, and popliteal lymph nodes), and spleen were isolated, mechanically disrupted, and filtered through a 40-μm stainless steel mesh to obtain single-cell suspensions. Bone marrow cells were recovered by placing the bone into a pierced 0.5-mL Eppendorf tube containing 200 μL of sterile 1× PBS. The tube was then placed in a 1.5-mL Eppendorf tube, and spun at 5000 rmp for 5 min to recover the cell suspension. This suspension was finally passed through a 70-micron filter (QTY1 FALCON Cell Strainer Filter).

## Flow cytometry staining

Single-cell suspensions were incubated with anti-CD16/32 (2.4G2; BD) to block Fc receptors before staining with PE, allophycocyanin, CD1d-α-GalCer-loaded tetramer, as previously described[54]. Binding was revealed with fluorochrome-conjugated Abs diluted in FACS buffer (PBS containing 5% FCS and 0.02% sodium azide). Antibodies are listed in supplemental material and methods. For intranuclear staining, cells were fixed and permeabilized after staining, then the Transcription Factor Buffer Set (BD Biosciences) was used according to the manufacturer's instructions. For intracellular cytokine staining, cells were fixed with 2% paraformaldehyde (Sigma-Aldrich) and permeabilized with saponin. The Zombie NIR Fixable Viability staining kit (Biolegend) was used according to the protocol provided. Flow cytometry was performed on a four-laser LSR Fortessa cytometer (BD Biosciences), data were collected with FACS Diva software V8 (BD Biosciences), and analyzed using FlowJo V10.7.2 (BD Biosciences).

## Generation of BM chimeras

BM cells were harvested from both the femurs and tibias of 3–6-week-old donors. B6 and BALB/c BM cells (5 × 10$^6$ in total) were intravenously transferred into lethally (9 Gray) irradiated 6–8-week-old F1: B6 CD1d1−/−CD1d2−/− × BALB/c CD1d1−/−CD1d2−/− mice. Cells were injected into the lateral tail vein within 24 h, either alone or co-injected at a 1:1 ratio. BM chimeras were analyzed at 8–10 weeks post-reconstitution. F1 mice were used as recipients to avoid potential rejection of the BM cells and to allow B6- and BALB-derived cells to be distinguished.

## In vitro stimulation

To measure intracellular cytokine production, cells were stimulated with 50 ng/mL PMA and 3300 ng/mL ionomycin in the presence of 5 mg/mL brefeldin A for 4 h (all reagents from Sigma-Aldrich).

## Quantitative real-time PCR

Pre-selection DP thymocytes (CD69 − CD4 + CD8+) were sorted using a FACS Aria III (BD Biosciences). Total mRNA was extracted from sorted cells−from BALB/c, BALB/c CD1d1−/− (expressing CD1d2), BALB/c CD1d2−/− (expressing CD1d1), C57BL/6 (expressing CD1d1, CD1d2 is a pseudogene), and C57BL/6 CD1d1−/− (expressing CD1d2)−using Direct-zol DNA/RNA Miniprep kit (Zymo Research) as recommended by the manufacturer. Reverse transcription was carried out using the TaqMan Fast Virus 1-Step Master Mix (ThermoFisher) in line with the manufacturer's instructions. Quantitative real-time PCR was performed on a 7500 Fast Real-Time PCR System (Applied Biosystems) using TaqMan gene expression assays for Gapdh (assay number: Mm99999915_g1), CD1d1 (assay number: Mm00783541_s1), and CD1d2 (assay number: Mm00776138_gh). Relative expression of CD1d1 and CD1d2 was determined by the comparative Ct method, using Gapdh as the internal control.

## Confocal microscopy and image analysis

A total of 20-μm-thick sections of frozen BALB/c and B6 4get thymuses embedded in OCT (Sakura Finetek) were stained with Alexa Fluor 647 conjugated monoclonal anti-Nrp2 antibody (1:80, Clone C-9; Santa Cruz Biotechnologies), counterstained with 1 μg/ml DAPI and mounted with Mowiol fluorescent mounting medium (Calbiochem), as described in[55]. Images were acquired with a LSM 780 confocal microscope (Carl Zeiss Microscopy). Confocal microscopy images stained for Nrp2 were segmented and quantified using ImageJ. Images were sequentially treated as follows: (i) median filtering over 0.5 pixels to reduce noise, (ii) thresholding, using the automated Intermodes method (implemented in ImageJ) to detect labelled cells, (iii) measurement of the mean gray value of each cell with a size greater than 10 pixels[2]. Data were collected with Zen Blue 3.5 software (Zeiss) and analyzed with MATLAB V2023b (MathWorks).

## Statistical analysis

FACS data are summarized as mean + SEM. The statistical significance of differences between populations was assessed based on a non-parametric Mann Whitney U test performed using GraphPad Prism software 8.0; $p$ values < 0.05 were considered significant.

## Reporting summary

Further information on research design is available in the Nature Portfolio Reporting Summary linked to this article.

# Data availability

Data supporting the findings of this study are provided either in supplementary figures or in the Source data file. Source data are provided with this paper.

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

## Acknowledgements

We thank staff at the IRSL genomic and FACS facility, particularly Christelle Doliger and Sophie Duchez, and at the institute's animal facility, particularly Veronique Parietti. We are indebted to the NIH Tetramer Facility for providing CD1d-tetramers; to David Voehringer (Universität Erlangen-Nürnberg, Germany) for providing us 4get mice and to Lionel Le Bourhis, Pablo Pereira and David Garrick for helpful discussions. The work reported in this paper was financially supported by INSERM, idex-NKT-diff (FP IDEX UP RM27J21IDXE3-NKTDIFF) (K.B.), Université Paris Diderot, Ministère de l'Enseignement supérieur, de la Recherche et de l'Innovation (L.A, and LA.FM), ANR RANKL (ANR-19-CE18-00021-01) (M.I.) and ANR Hu-Thy-L (ANR-21-CE15-0008-01) (A.T.). A.T. and E.C. are supported by the French Government's Investissement d'Avenir Program, Laboratoire d'Excellence "Milieu Intérieur" (Grant ANR-10-LABX-69-01). INSERM UMR 1160 is a member of OPALE Carnot Institute, The Organization for Partnerships in Leukemia, Institut de Recherche Saint-Louis, Hôpital Saint-Louis, Paris, France (www.opale.org). Schematics were created with the help of BioRender.com. In memory of late Jean-Claude Brouet and Albert Bendelac.

## Author contributions

L.A. designed, performed, and analyzed the experiments; L.A.F.M., G.C., A.S., and M.I. performed and analyzed experiments; R.P. and M.D.C. performed experiments and provided essential tools and knowledge; C.K., M.D., C.V., J.J., C.L., S.K., JC.M., H.G., E.C., A.T., and B.L. analyzed experiments, and provided essential tools and/or knowledge; J.K. designed and analyzed experiments, partly supervised the study, and provided essential tools and knowledge; K.B. designed and analyzed experiments, wrote the paper and supervised the study.

## Competing interests

The authors declare no competing interests.
