## [Peer Review File · Nature Communications]

Intrinsic factors and CD1d1 but not CD1d2 expression levels control invariant natural killer T (NKT) cell subset differentiationREVIEWER COMMENTS

Reviewer #1 (Remarks to the Author):

There are two closely linked genes encoding CD1d, and both have the capability to select NKT1, NKT2, and NKT17 - subsets in the mouse thymus. The authors have tested the proposition that the increased expression of the minor CD1d gene in BALB/c mice, encoded by Cd1d2, is responsible for the increased NKT2/NKT17 cells in this strain. The answer is negative, CD1d2 protein is not highly expressed, in agreement with previous studies, and it does not greatly influence NKT cell subsets. Instead, they show that NKT cell-intrinsic mechanisms are responsible. They also studied CD1d^{+/-} heterozygous mice and showed that in this context, the amount of CD1d expression influences NKT cell differentiation, providing evidence that decreased signal strength favors the NKT1 population. This conclusion is confirmatory of several previous studies, and there is little further insight into the mechanism for thymic NKT cell subset prevalence from this work. Why do BALB mice have more NKT2/17 despite lower CD1d expression? However, the authors have gone further than the previous experiments to prove their points, and they have created the appropriate genetic models for testing CD1d1 and CD1d2 function. Furthermore, the experiment in Fig. 4 indicates that not only is the level of CD1d expression not governing for determining the NKT cell subset frequency but also that B6 and BALB positively selecting double positive thymocytes are not presenting different ligands that induce differential subset representations.

Why do the authors refer to CD1d2 in B6 as a pseudogene? Their data and previous research (ref 29) show that B6 CD1d2 can select NKT cells.

The data showing increased Vb7 in CD1d^{+/-} mice was shown originally by MacDonald (PMID 16455960).

Fig. 1B and elsewhere. Are the NKT cells in BALB/c mice larger in size? Should the MFI be corrected for a cell size difference?

Some of the flow cytometry measurements indicate populations have a bi-modal staining pattern. Some examples include Fig S1A PLZF staining for B6 NKT2 and Fig S1B SLAMF6 staining for some BALB/c and B6 subsets. Do the authors have an explanation or speculation?

Line 439: "In fact, the subsets skewed toward Vb7 expression have a developmental advantage (NKT2 or NKT1), whereas the subset with the lowest Vb7 expression (NKT17) has a reduced developmental potential." How was developmental potential measured?

Reviewer #2 (Remarks to the Author):

In this ms, Amable and colleagues investigated the cues that drive NKT differentiation into NKT1, NKT2, and NKT17 subsets after agonist selection in the thymus. This is an exciting and complete piece of work and should be suitable for the readership of natcomms.

The authors thoroughly investigated the role of CD1d1 and CD1d2 isogenes for thymic NKT differentiation and concluded that CD1d2 expression in Balb/c but not in B6 is not the reason for an NKT2 bias in Balb/c. To this end, they generated individual KO mice for the two CD1d isoforms. Strong evidence for rather CD1d-unrelated intrinsic factors driving the distinct differentiation of B6 vs. Balb/c NKT cells is presented in mixed BM Xmera in Fig.4. Next, they use elegant haploinsufficient mice to show that CD1d1 expression levels still influence the differentiation of NKT0 into NKT17 cells

I would suggest that the title ("Intrinsic factors control ...") is maybe a bit misleading and

could be adapted, as the data shown indicate that rather “Intrinsic factors and CD1d1 but not CD1d2 expression levels control NKT differentiation into NKT1, NKT2, and NKT17 subsets”. Also, it would be nice to discuss candidates for such intrinsic factors that favor NKT2 in Balb/c. Do these factors also promote type-2 innate MAIT and gd T cell development in Balb/c or is this specific for NKT differentiation?
Very minor: The call-out for Fig 5f is missing and Fig 5g is mentioned after Fig 6.

We thank both reviewers for their constructive comments.

Reviewer #1 (Remarks to the Author):

Why do the authors refer to CD1d2 in B6 as a pseudogene? Their data and previous research (ref 29) show that B6 CD1d2 can select NKT cells.

In WT B6 mouse strain, we refer to CD1d2 as a pseudogene because in this strain the gene harbors a frameshift mutation at the beginning of the fourth exon encoding the a3 domain of CD1D2. This mutation is predicted to abolish its surface expression (*The Journal of Immunology*, 1998, 160: 3128–3134). *Cd1d2* is therefore considered a pseudogene in B6 WT mice. This frame shift mutation does not exist in 129/sv and BALB/C strains, where *Cd1d2* is not a pseudogene. We agree with the referee that in our study and in the study by Gapin's group (ref 29), the results indicate that B6 CD1d2 can nevertheless select NKT cells. However, it should be kept in mind that these mice are CD1d1-deficient mice expressing CD1d2 as they were generated with ES cells from the 129/SV mouse strain expressing CD1d2.

The data showing increased Vb7 in CD1d+/- mice was shown originally by MacDonald (PMID 16455960).

The referee is correct, and we have now mentioned in the discussion that our results are in agreement with those of MacDonald's study. In addition, our results show that the increase in Vb7 levels differs between NKT cell subsets (Fig. 5E).

Fig. 1B and elsewhere. Are the NKT cells in BALB/c mice larger in size? Should the MFI be corrected for a cell size difference?

NKT cells in BALB/c mice are not larger in size and thus there is no need to correct MFI for cell size. We have added this information to supplemental figure S1 (S1C).

Some of the flow cytometry measurements indicate populations have a bi-modal staining pattern. Some examples include Fig S1A PLZF staining for B6 NKT2 and Fig S1B SLAMF6 staining for some BALB/c and B6 subsets. Do the authors have an explanation or speculation?

Baranek et al. (*Cell Reports* 32, 108116, September 8, 2020) performed an unbiased single-cell transcriptomic analysis (scRNA-seq) on thymic NKT cells. This study revealed discrete subsets among NKT1 (NKT1a, b, and c) and NKT2 (NKT2a, and b). It also showed that NKT2 cells comprise precursors to NKT1 and NKT17 cells. We thus speculate that the bi-modal staining pattern observed with our cells could be due to the heterogeneity of NKT cell subsets.

Line 439: "In fact, the subsets skewed toward Vb7 expression have a developmental advantage (NKT2 or NKT1), whereas the subset with the lowest Vb7 expression (NKT17) has a reduced developmental potential." How was developmental potential measured?

We are sorry about the confusion induced by this passage, as we did not measure any developmental potential. The sentence has now been replaced by the following one in line 339: "Our results show that V β 7 use correlates with changes in NKT2 and NKT17 but not NKT1 cell frequencies in CD1d $^{+/-}$ mice. In fact, the subsets skewed toward V β 7 expression have an unaltered (NKT1) or increased (NKT2) frequency, whereas the subset with the lowest V β 7 expression (NKT17) has a reduced cell frequency".

Reviewer #2 (Remarks to the Author):

I would suggest that the title ("Intrinsic factors control ...") is maybe a bit misleading and could be adapted, as the data shown indicate that rather "Intrinsic factors and CD1d1 but not CD1d2 expression levels control NKT differentiation into NKT1, NKT2, and NKT17 subsets".

We took this advice into consideration and changed the title as suggested.

Also, it would be nice to discuss candidates for such intrinsic factors that favor NKT2 in Balb/c

We have answered this question based on newly obtained results that we correlated with data from the literature. Below is a short version of the detailed discussion from lines 434 to 503, and in supplemental figure S5.

To address this question, we analyzed the expression of intrinsic factors related to the development and/or function of NKT1 (T-bet, IL-15R), NKT2 (IL-25R), and NKT17 (ROR γ t, IL-7R, TGF β RII, phospho-SMAD2/3). We found no differences in their expression levels that could explain differential distribution of NKT cells in B6 vs BALB/c mice (data provided below in Figure 1 to the reviewers). However, we found that GATA3 was expressed at two-fold higher levels in NKT2 cells from BALB/c compared to cells from B6 mice (Supp Fig. S5A). Similarly, our results in figure Fig.1C and Supp Fig. S1D reveal higher expression of SLAM proteins at the surface of NKT2 cells from BALB/c compared to B6 mice. Both GATA3 and SLAM-associated proteins (SAP) have been described to favor NKT2 cell development (*J Immunol*, 2006, 177 : 6650-6659 ; *Eur J Immunol*, 2016, 46: 2162-2174), their higher expression in BALB/c NKT2 cells could contribute in part to the predominance of this subset in these mice.

In addition, we exploited a dataset made available by Georgiev et al. (*Nat Commun*, 2016, 7 : 13116) following their analysis of transcriptional expression of NKT cell subsets in B6 vs BALB/c mice. We found EZH2 transcript expression to be higher in B6 NKT2 cells compared to BALB/c NKT cells (Supp Fig. S5B). Our intracellular staining analysis showed a higher H3K27me3 level in B6 NKT2 cells compared to BALB/c NKT cells (Supp Fig. S5C). Based on a study showing that loss of Ezh2 and H3K27me3 in NKT cells favored NKT2 cell development (*J Exp Med*, 2015, 212 : 297-306), its reduced level in BALB/c NKT cells could contribute in part to the higher NKT2 cell frequency in BALB/c mice.

By further exploiting the Georgiev et al. dataset (*Nat Commun*, 2016, 7 : 13116), we found NRP2 transcripts (encoding for neuropilin-2) to be highly expressed in BALB/c NKT2 cells compared to B6 mice (Supp Fig. S5D). We confirmed this increased expression by immunohistochemistry using an

IL-4/GFP-enhanced transcript (4Get) strain, which expresses a fluorescent reporter in IL-4-producing cells, to visualize NKT2 cells (Supp Fig. 5E). NRP2 mainly binds to semaphorin 3F and 3C (Sema3F, 3C) (*Front Immunol*, 2017, 8 :1228). In humans, the NRP2/Sema3F axis is reported to inhibit thymocyte migration in response to S1P1, a chemokine with a well-documented role in thymocyte egress from the thymus (*PLoS One*, 2014, e103405). Because BALB/c NKT2 cells express NRP2, this may render emigration of NKT2 inefficient in this strain, providing a potential explanation as to why NKT2 cells are more frequent in BALB/c mice. Future studies using NRP2-deficient mice will allow the functional consequences of NRP2 expression in BALB/c NKT2 cells to be verified.

Based on these considerations, our results thus indicate that the intrinsic factors involved in the differential strain subset distribution are likely to be multiple, including transcription factors, downstream signaling cascades, epigenetic factors, and migratory factors, among others. The fine regulation of these intrinsic factors can be linked to gene polymorphisms and/or regulatory enhancer/silencer regions. To uncover new genes controlling NKT cell differentiation and strain distribution, we are planning to deploy an unbiased genetic approach, using a collection of mice (CC collection) (*Nat Genet*, 2004, 36 :1133-1137). This collection will allow us to correlate NKT cell subset distribution with genetic controls. Preliminary results show a variability in the distribution of NKT2 cells in the 8 founders and the 35 strains of the CC collection analyzed so far, supporting the use of this approach to search for genetic factors controlling NKT cell subset differentiation and distribution (data not shown).

Do these factors also promote type-2 innate MAIT and $\gamma\delta$ T cell development in Balb/c or is this specific for NKT differentiation?

These factors could potentially also promote type-2 MAIT and $\gamma\delta$ T cell development in BALB/c mice. In fact, in addition to being important for NKT2 cells development (*J Immunol*, 2006, 177 : 6650-6659), GATA3 is also important for $\gamma\delta$ 2 development; these cells were efficiently depleted in Gata3 cKO mice (*Nature communications*, 2020, 11 : 18155-8). Within the $\gamma\delta$ T cell compartment, it was also previously demonstrated that development of the $\gamma\delta$ 2 V γ 1V δ 6.3 ($\gamma\delta$ NKT) subset requires SAP (*Journal of Immunology*, 2010, 84: 6746–6755), indicating that SLAM/SAP signaling during development contributes to the emergence of NKT cells, MAIT cells, and at least the $\gamma\delta$ 2 T cell subset.

Very minor: The call-out for Fig 5f is missing and Fig 5g is mentioned after Fig 6.

We have corrected the order of appearance of the figures in the text

Figure 1: Expression of intrinsic factors related to the development and/or function of NKT1 (T-bet, IL-15R (CD122)), NKT2 (IL-25R (IL-17RB)), and NKT17 (ROR γ t, IL-7R, TGF β RII, phospho-SMAD2/3) in NKT cell subsets of B6 and BALB/c mice. Individual/ average MFI for these markers are shown. Data are representative of 2 to 7 experiments in 7-8-week-old mice, n=4-6 animals per strain in each experiment. Nonparametric Mann Whitney U test was used. NS not significant ($P > 0.05$).

REVIEWER COMMENTS

Reviewer #1 (Remarks to the Author):

This revised manuscript described experiments designed to address why the functional subsets of thymic natural killer T (NKT) cells differ greatly between B6 mice, which have mostly Th1-like or NKT1 cells, compared to BALB/c, with more NKT2 cells. The manuscript is improved in some respects by the author's response, and it establishes that NKT cell precursor intrinsic factors are responsible. They have added new data showing increased expression of GATA-3 in BALB/c, and decreased Ezh2 in BALB/c NKT thymus subsets. Based on B6 strain genetic alterations, these were predicted to increase NKT2 cells. They also found increased Nrp2 expression in BALB/c, based on data mining, but did not do experiments to validate its importance. This section belongs in the results, and it is poorly written, with several references to "data not shown," which is no longer acceptable for most journals, and discussion of experiments they might do in the future. The antibody staining for Nrp2 is not properly described in the text, and the increase in H3K27me3 in BALB/c NKT cells does not agree with what is written.

Some data from other labs and data presented here suggest that decreased TCR affinity during selection will promote NKT1 (or conversely, that NKT2 cells have obtained an increased TCR signal). If this is the case, then why don't CD1d^{+/-} mice, which presumably have a decreased signal, also have fewer NKT2 cells? Sorry that this point was not raised earlier by this reviewer.

Minor issues

Use of scientific notation for the numbers in Fig 1A

Line 139. Is the CD1d2 TG mice on the B6 or CD1d1^{-/-} background?

Line 422 needs a slight clarification to note that while NKT1 cells are increased in the BALB/c chimeras compared to BALB/c mice, there is still a higher frequency of NKT2 in the chimeras than in B6 mice.

Reviewer #2 (Remarks to the Author):

The authors did a great job in thoroughly replying to all the two reviewers' concerns. The ms should now be very suitable for publication in NCOMMS.

However, I am not sure if "data not shown" on page 21 is supported by the journal's policy.

We thank both reviewers for their constructive comments.

Reviewer #1 (Remarks to the Author):

This revised manuscript described experiments designed to address why the functional subsets of thymic natural killer T (NKT) cells differ greatly between B6 mice, which have mostly Th1-like or NKT1 cells, compared to BALB/c, with more NKT2 cells. The manuscript is improved in some respects by the author's response, and it establishes that NKT cell precursor intrinsic factors are responsible. They have added new data showing increased expression of GATA-3 in BALB/c, and decreased Ezh2 in BALB/c NKT thymus subsets. Based on B6 strain genetic alterations, these were predicted to increase NKT2 cells. They also found increased Nrp2 expression in BALB/c, based on data mining, but did not do experiments to validate its importance. This section belongs in the results, and it is poorly written, with several references to "data not shown," which is no longer acceptable for most journals, and discussion of experiments they might do in the future. The antibody staining for Nrp2 is not properly described in the text, and the increase in H3K27me3 in BALB/c NKT cells does not agree with what is written.

We thank the referee for all remarks allowing to improve our manuscript.

- **The section is now moved to the result as an independent paragraph titled « Multiple NKT cell precursor intrinsic factors are likely to be involved in differential strain-related subset distribution », starting line 311, associated to Fig. 5.**
- **We provided the relevant data for the « data not show ». Please, find at the end a list of what data have been added and where it was introduced.**
- **We have now properly described in the text the antibody staining for Nrp2 (line 652).**
- **We have corrected the color code in the H3K27me3 histogram. The increase in H3K27me3 in BALB/c NKT (Fig. 5C) cells does agree now with what is written (line 347). Sorry for this confusion.**

Some data from other labs and data presented here suggest that decreased TCR affinity during selection will promote NKT1 (or conversely, that NKT2 cells have obtained an increased TCR signal). If this is the case, then why don't CD1d^{+/-} mice, which presumably have a decreased signal, also have fewer NKT2 cells? Sorry that this point was not raised earlier by this reviewer.

We thank the referee for this pertinent question. In fact, data from other laboratories suggest that decreased TCR signal during selection will promote NKT1 cells and conversely disfavor NKT2 cells (ref 17, and 23). CD1d^{+/-} mice, which have a decreased signal, do not however have fewer NKT2 cells. This is likely due to that fact that the defect in TCR signal in these mice is extrinsic to NKT cells (reduced CD1d expression in selecting DP cells) contrary to previous studies where mouse models expressing hypomorphic alleles of Zap70 were studied (ref 17, and 23). In CD1^{+/-} mice, where components of TCR signaling cascade are not altered, NKT2 cells likely adapt to the reduced CD1d

expression by increased usage of V β 7 that confer to these cells a higher affinity for ligand and allows their increased frequency. This paragraph is added in the discussion line 523.

Minor issues

Use of scientific notation for the numbers in Fig 1A

We used scientific notation for numbers in Fig1A and applied this for other figures.

Line 139. Is the CD1d2 TG mice on the B6 or CD1d1^{-/-} background?

The CD1d2 TG mice is on the B6 CD1d1^{-/-} background ? this has been changed in the text now line 138.

Line 422 needs a slight clarification to note that while NKT1 cells are increased in the BALB/c chimeras compared to BALB/c mice, there is still a higher frequency of NKT2 in the chimeras than in B6 mice.

The referee is right, the sentence « This profile is characteristic of that observed in B6 mice » is confusing as it stands for NKT1 but not NKT2 cells. We took it away from the text (no longer in line 479, former 422).

Reviewer #2 (Remarks to the Author):

The authors did a great job in thoroughly replying to all the two reviewers' concerns. The ms should now be very suitable for publication in NCOMMS.

However, I am not sure if "data not shown" on page 21 is supported by the journal's policy.

We thank the referee for his advice that helped improve our manuscript.

We provided the relevant data for the « data not shown ». Please, find below a list of what data have been added and where it was introduced.

The relevant data for the « data not shown » are :

1/ thymic NKT cell absolute numbers of BALB/c strains, added in figure 2B (line 217).

2/ thymic NKT cell absolute numbers of B6 strains, added in figure 2E (line 226).

3/ Expression levels of intrinsic factors related to the development and/or function of NKT1 (T-bet, IL-15R), NKT2 (IL-17RB: IL-25R), and NKT17 (ROR γ t, IL-7R, TGF β RII, phospho-SMAD2/3), added as Supplemental Figure S3 (line 319).

4/ representative staining for CD4 and CD8 expression in thymocytes from CD1^{+/+} and CD1d^{+/-} mice, added in Fig. 6B, as Fig. 6B upper panel (line 378).

5/ thymic NKT cell absolute numbers in CD1^{+/+} and CD1^{-/-} mice, added in figure 6C (line 381).

6/ NKT cell subset distribution in B6 \rightarrow B6 J α 18^{-/-} bone marrow chimeras, added as Supplemental figure S6 (line 484).

REVIEWERS' COMMENTS

Reviewer #1 (Remarks to the Author):

The authors have made a thorough response to the critique

Reviewer #2 (Remarks to the Author):

The "data not shown" issues were clarified in the revised version.

We thank both reviewers for their constructive comments that improved our manuscript.

REVIEWERS' COMMENTS

Reviewer #1 (Remarks to the Author):

The authors have made a thorough response to the critique

We are pleased that our response satisfied the reviewer

Reviewer #2 (Remarks to the Author):

The "data not shown" issues were clarified in the revised version.

We are pleased that our clarification satisfied the reviewer